

# Extensive long-range entanglement
# in a nonequilibrium steady state

Shachar Fraenkel⋆ and Moshe Goldstein

Raymond and Beverly Sackler School of Physics and Astronomy,
Tel Aviv University, Tel Aviv 6997801, Israel

⋆ shacharf@mail.tau.ac.il

## Abstract

Entanglement measures constitute powerful tools in the quantitative description of quantum many-body systems out of equilibrium. We study entanglement in the current-carrying steady state of a paradigmatic one-dimensional model of noninteracting fermions at zero temperature in the presence of a scatterer. We show that disjoint intervals located on opposite sides of the scatterer, and within similar distances from it, maintain volume-law entanglement regardless of their separation, as measured by their fermionic negativity and coherent information. The mutual information of the intervals, which quantifies the total correlations between them, follows a similar scaling. Interestingly, this scaling entails in particular that if the position of one of the intervals is kept fixed, then the correlation measures depend non-monotonically on the distance between the intervals. By deriving exact expressions for the extensive terms of these quantities, we prove their simple functional dependence on the scattering probabilities, and demonstrate that the strong long-range entanglement is generated by the coherence between the transmitted and reflected parts of propagating particles within the bias-voltage window. The generality and simplicity of the model suggest that this behavior should characterize a large class of nonequilibrium steady states.

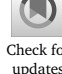

# 1  Introduction

Within the broad field of quantum many-body physics, the study of nonequilibrium phenomena is becoming increasingly intertwined with the analysis of entanglement witnesses. In particular, the scaling of various entanglement measures with the size of a subsystem quantitatively captures canonical nonequilibrium behaviors, such as thermalization [1–3] or the violation thereof [4–6], in closed systems subjected to an initial quench. In quench problems of this type, transient effects of long-range entanglement are signatures of integrability [7–10], and the dynamics as well as the stationary values of the entanglement entropy, negativity, and mutual information are used for the classification of out-of-equilibrium models and their phases [11–22].

This success motivates the examination of entanglement properties also in open systems, and specifically those of their steady states, which may give rise to unique entanglement structures [23–29]. Current-carrying states of inhomogeneous systems offer a promising ground for such an analysis, as recent studies have revealed that they can naturally sustain long-range quantum coherent correlations [30–32]. In this context, scaling laws of steady-state entanglement measures were shown to be closely related to the localized-diffusive phase transition of the open noninteracting Anderson model [33]. In this work we show that nonequilibrium conditions may lead to an even more striking behavior of quantum information measures. This is achieved through the study of an elementary model for an inhomogeneous system in a current-carrying state, where the mechanism underlying its unusual entanglement properties is exceptionally transparent.

Beyond the role of entanglement measures as fundamental quantities, their estimation is inextricably linked to the construction of useful tensor-network simulations of condensed matter systems [34, 35]. Strong (volume-law) entanglement, which is commonly found in nonequilibrium quantum many-body states [2, 13, 36], impedes the utility of these simulation methods [37]. One possible key for their improvement is thus the uncovering of nontrivial entanglement structures in certain classes of states, like the one that is the subject of this work. Steady states that are predicted to give rise to strong entanglement are also of potential interest from a technological standpoint, as entanglement is an essential resource for quantum information applications [38–40].

In this work, we examine the long-range entanglement induced by a current-conserving scatterer in the voltage-biased steady state of a 1D noninteracting fermion system. We treat this problem generally, without imposing specific assumptions regarding the structure of the

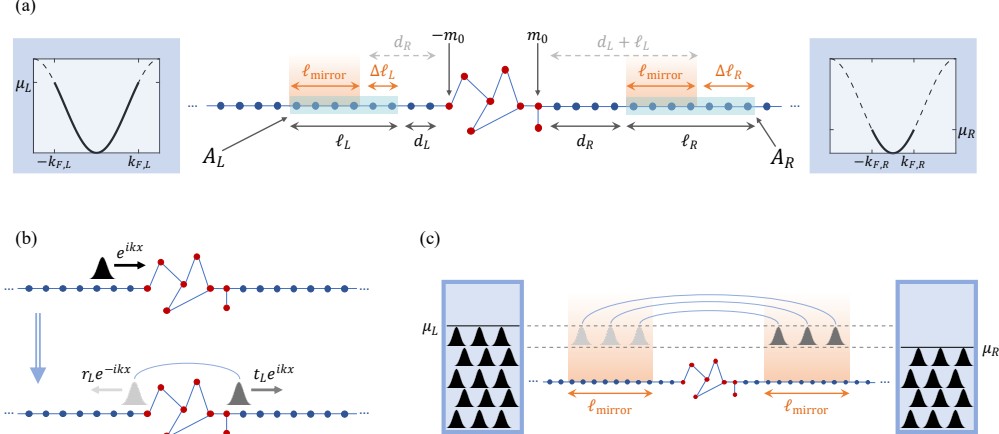

Figure 1: (a) Schematic sketch of the model: Red circles mark lattice sites in the scattering region, while sites outside this region are marked in blue. Noninteracting reservoirs with different chemical potentials are connected to the two ends of the chain. See the text (Sec. 2) for details regarding the notations. Bottom panels: An intuitive picture for the origin of the steady-state entanglement structure. (b) Any incoming wavepacket (black) is split by the scattering region into a transmitted part (dark gray) and a reflected part (light gray), with amplitudes determined by the associated scattering matrix (see Eq. (5)). The transmitted and reflected parts are coherently correlated and thus generate entanglement. (c) Split wavepackets with energies within the voltage window strongly entangle regions that mirror each other with respect to the position of the scattering region. Correlation measures exhibit long-range volume-law scaling, since the number of split wavepackets shared by these mirroring regions is proportional to their length and independent of their spatial separation.

scatterer other than it being smaller compared to all other length scales. We study the correlations between two disjoint subsystems located on opposite sides of the scatterer: $A_L$ on its left, and $A_R$ on its right. The total amount of correlations is regularly quantified using the mutual information (MI) between the two subsystems,

$$\mathcal{I}_{A_L:A_R} = \mathcal{S}_{A_L} + \mathcal{S}_{A_R} - \mathcal{S}_A. \tag{1}$$

Here $A = A_L \cup A_R$, and $\mathcal{S}_X = -\text{Tr}[\rho_X \ln \rho_X]$ is the von Neumann entanglement entropy of a subsystem $X$ [41], with $\rho_X$ being the reduced density matrix of $X$.

Given that $A$ is in a mixed state, however, the MI has limitations as a measure of entanglement, since it takes into account both classical and quantum correlations [42]. Therefore, we also address the fermionic negativity [43–45] between $A_L$ and $A_R$, an entanglement monotone defined as

$$\mathcal{E} = \ln \text{Tr} \sqrt{(\widetilde{\rho}_A)^{\dagger} \widetilde{\rho}_A}, \tag{2}$$

where $\widetilde{\rho}_A$ is obtained from $\rho_A$ by applying a partial time-reversal to either $A_L$ or $A_R$. Interestingly, our analysis shows that the MI and negativity follow a similar scaling, a scaling which to the best of our knowledge has not been previously observed in a natural physical scenario.

As our main result, we find that both the MI and the negativity scale linearly with $\ell_{\text{mirror}}$, the number of sites in $A_L$ that, under reflection with respect to the position of the scatterer, overlap with sites in $A_R$ (see Fig. 1(a) for an illustration). Remarkably, this steady-state extensive entanglement is long-ranged, as the observed volume-law scaling does not decay with the (similar) distance of the mirroring sites from the scatterer. Moreover, the

entanglement depends non-monotonically on the distance of either $A_L$ or $A_R$ from the scatterer. We analytically derive exact formulas for the asymptotic scaling of the MI and the negativity (Eqs. (8)–(9)). Additionally, we demonstrate that the coherent information (CI) [46, 47],

$$I(A_L\rangle A_R) = \mathcal{S}_{A_R} - \mathcal{S}_A, \tag{3}$$

is not only positive (which is impossible classically) when $\ell_{\text{mirror}}$ is large enough, but also grows with $\ell_{\text{mirror}}$ according to a volume law (Eq. (10)).[1] The CI is a lower bound to the squashed entanglement [48, 49], another rigorous entanglement measure with axiomatically desirable properties [41], which therefore obeys an extensive scaling as well in regimes where $I(A_L\rangle A_R) > 0$.

A simple intuitive explanation for these results is provided by considering that the scatterer splits each incoming single-particle wavepacket into two coherently correlated counter-propagating parts (Fig. 1(b)). Each such split wavepacket with energy within the voltage window generates entanglement, since detecting the particle in one subsystem prohibits its presence in the other. As the number of such wavepackets is proportional to $\ell_{\text{mirror}}$ and independent of the distance between the subsystems (Fig. 1(c)), the correlation measures exhibit a similar behavior.

The paper is organized as follows. In Sec. 2 we introduce the model for the system and its nonequilibrium steady state that are the subject of this work. In Sec. 3 we report our analytical results for correlation measures in the steady state. We point out the salient features of these results, and support them through comparisons to numerical results (computed for a specific choice of the scatterer). Sec. 4 outlines the derivation of the analytical results, and is limited to the conceptually crucial steps in the derivation, while the technical aspects of the process are mostly discussed in the appendices. In Sec. 5 we conclude and mention potential future directions arising from this work.

Additionally, the paper includes four technical appendices. In Appendix A we derive the two-point correlation function, which served as the basic ingredient in all of our calculations. Appendix B presents the technical details of our computation method for subsystem entropies, from which (as explained in Sec. 4) the asymptotics of the MI and CI can be immediately derived. Appendix C summarizes the derivation of the formula for the fermionic negativity, which is based on the same method. Finally, in Appendix D we complement the numerical results included in Sec. 3 with additional numerical tests corroborating our analysis.

## 2 Nonequilibrium model

We consider a 1D lattice, occupied by noninteracting fermions and connected at its ends to two reservoirs with different chemical potentials, $\mu_L \neq \mu_R$, at zero temperature. The lattice is assumed to be of infinite length, and it is modeled as a tight-binding chain that is homogeneous everywhere, except for a small region at the center of the chain, which we dub *the scattering region*. The Hamiltonian is thus of the form

$$\mathcal{H} = -\eta \sum_{m=m_0}^{\infty} \left[ c_m^\dagger c_{m+1} + c_{-m}^\dagger c_{-m-1} + \text{h.c.} \right] + \mathcal{H}_{\text{scat}}. \tag{4}$$

Here $c_m$ is a fermionic annihilation operator for the $m$th lattice site, $\eta > 0$ is a hopping amplitude, $m = \pm m_0$ designate the boundaries of the scattering region, and $\mathcal{H}_{\text{scat}}$ pertains only to sites with $|m| \leq m_0$ and breaks the homogeneity, e.g., through modified hopping terms, on-site energies, or side-attached sites.

---

[1]The definition of the CI is evidently not symmetric with respect to the two subsystems $A_L$ and $A_R$. Our choice to examine $I(A_L\rangle A_R)$ rather than $I(A_R\rangle A_L)$ is arbitrary, and we maintain this choice throughout the text for convenience.

The scattering region can be associated with a $2 \times 2$ unitary scattering matrix [50], defined for any lattice momentum $0 < k < \pi$:

$$S(k) = \begin{pmatrix} r_L(k) & t_R(k) \\ t_L(k) & r_R(k) \end{pmatrix}. \tag{5}$$

The diagonal (off-diagonal) entries of this matrix stand for reflection (transmission) amplitudes; the left (right) column contains the scattering amplitudes for a particle originating in the left (right) reservoir with momentum $k > 0$ $(-k < 0)$. The squared moduli of the entries correspond to the transmission and reflection probabilities, respectively $\mathcal{T}(|k|)$ and $\mathcal{R}(|k|) = 1 - \mathcal{T}(|k|)$, for a particle originating in either reservoir with momentum $k$. These scattering probabilities are the sole property of the scatterer on which our analytical results depend.

The single-particle eigenbasis of the Hamiltonian is comprised of extended scattering states with energies $\varepsilon = -2\eta \cos k$, and of bound states localized near the scattering region [50,51]; we ignore the latter in our analysis, as they contribute negligibly to correlations between sites that are distant from the scatterer. The current-carrying many-body steady state is pure, with single-particle scattering states originating in the left (right) reservoir occupied up to a Fermi momentum $k_{F,L} > 0$ $(-k_{F,R} < 0)$, as shown schematically in Fig. 1(a). The Fermi momenta are related to the chemical potentials through $\mu_i = -2\eta \cos k_{F,i}$ $(i = L, R)$.

Correlation and entanglement measures are calculated with respect to two subsystems $A_L$ and $A_R$, each comprised of contiguous sites, with lengths $\ell_i$ and distances $d_i \geq 0$ $(i = L, R)$ from the scattering region (all of which are assumed to be much larger than the size of the scattering region, $2m_0 + 1$): $A_L$ contains the sites $m$ such that $-d_L - \ell_L \leq m + m_0 \leq -d_L - 1$, while $A_R$ contains the sites $m$ such that $d_R + 1 \leq m - m_0 \leq d_R + \ell_R$ (see Fig. 1(a)). $\ell_{\mathrm{mirror}} = \max\{\min\{d_L + \ell_L, d_R + \ell_R\} - \max\{d_L, d_R\}, 0\}$ denotes the number of mirroring pairs $(-m, m) \in A_L \times A_R$, and we also define $\Delta\ell_i = \ell_i - \ell_{\mathrm{mirror}}$.

## 3  Asymptotics of correlation measures

The leading behaviors of the MI and the negativity can be encapsulated by that of the Rényi MI, defined as

$$\mathcal{I}^{(n)}_{A_L:A_R} = S^{(n)}_{A_L} + S^{(n)}_{A_R} - S^{(n)}_A, \tag{6}$$

where $S^{(n)}_X = \frac{1}{1-n} \ln \mathrm{Tr}[(\rho_X)^n]$ are Rényi entropies (which converge to $\mathcal{S}_X$ as $n \to 1$). We report that, for the nonequilibrium steady state described above, the Rényi MI follows a volume-law scaling with $\ell_{\mathrm{mirror}}$,

$$\mathcal{I}^{(n)}_{A_L:A_R} \sim \frac{\ell_{\mathrm{mirror}}}{1-n} \int_{k_-}^{k_+} \frac{dk}{\pi} \ln[(\mathcal{T}(k))^n + (\mathcal{R}(k))^n] + \dots, \tag{7}$$

where $k_- = \min\{k_{F,L}, k_{F,R}\}$ and $k_+ = \max\{k_{F,L}, k_{F,R}\}$ are the two Fermi momenta that bound the voltage window. The ellipsis (which will be henceforth omitted) represents subleading terms, the dominant of which are logarithmic in the different length scales ($\ell_i$, $d_i$, and combinations thereof). Full exact expressions for these logarithmic terms can be obtained in the long-range limit $d_i/\ell_i \to \infty$ (with $d_L - d_R$ kept fixed) using methods related to the asymptotic calculation of Toeplitz determinants [52–55]; the results for these subleading corrections will be discussed in a separate publication [56].

The MI is related to the Rényi MI simply by its definition, through the equality $\mathcal{I}_{A_L:A_R} = \lim_{n\to 1} \mathcal{I}^{(n)}_{A_L:A_R}$, yielding the following asymptotics:

$$\mathcal{I}_{A_L:A_R} \sim \ell_{\text{mirror}} \int_{k_-}^{k_+} \frac{dk}{\pi} \left[ -\mathcal{T} \ln \mathcal{T} - \mathcal{R} \ln \mathcal{R} \right] . \tag{8}$$

The negativity, on the other hand, is not *a priori* directly related to the Rényi MI, yet our analysis shows that, at the leading (linear) order,

$$\mathcal{E} \sim \ell_{\text{mirror}} \int_{k_-}^{k_+} \frac{dk}{\pi} \ln \left[ \mathcal{T}^{1/2} + \mathcal{R}^{1/2} \right] \sim \frac{1}{2} \mathcal{I}^{(1/2)}_{A_L:A_R} . \tag{9}$$

The equality $\mathcal{I}^{(1/2)}_{A_L:A_R} = 2\mathcal{E}$ is known to arise in quenches of integrable systems [10]. Eqs. (8) and (9) state that, for a generic non-trivial scatterer (i.e., unless $\mathcal{T}(k) \in \{0, 1\}$ for all $k \in [k_-, k_+]$), the MI and negativity both exhibit extensive scaling with $\ell_{\text{mirror}}$. Additionally, we have found that the CI scales at the leading order as

$$I(A_L)A_R) \sim (\ell_{\text{mirror}} - \Delta\ell_L) \int_{k_-}^{k_+} \frac{dk}{2\pi} \left[ -\mathcal{T} \ln \mathcal{T} - \mathcal{R} \ln \mathcal{R} \right] , \tag{10}$$

and so it grows linearly with $\ell_{\text{mirror}}$ if $\Delta\ell_L$ is fixed. Crucially, the asymptotics in Eqs. (7)–(10) do not depend on the magnitudes of $d_i$, and they hold even when $d_i \gg \ell_i$. That is, the extensive entanglement is long-ranged, and it holds even for subsystems that are very distant relative to their lengths, but that still share mirroring sites. Eqs. (8)–(10) are the central results of this work.

The special symmetric case where $\ell_L = \ell_R = \ell$ and the subsystems are positioned symmetrically relative to the scatterer ($d_L = d_R$) is particularly illuminating with regard to the nature of the steady-state entanglement. In this case we have found that $\mathcal{S}_A$ scales sublinearly with $\ell$, i.e. $\lim_{\ell\to\infty} \mathcal{S}_A/\ell = 0$ (see Eq. (19)). The combined subsystem $A$ is therefore weakly entangled to the rest of the system, while its two components – one being the mirror image of the other – maintain strong entanglement between them.

The volume-law terms in Eqs. (7)–(10) are evidently generated by the scattering states within the voltage window, with the contribution of each state in Eq. (7) being the equivalent of the statistical moment of its corresponding transmission probability. This simple form allows to deduce that the source of the long-range entanglement is the coherence between the reflected part and the transmitted part of each scattered particle, which arrive simultaneously at mirroring sites. In the steady state, the constant particle current renders this strong entanglement a stationary property, and the length scale $\ell_{\text{mirror}}$ determines the amount of entanglement as it is proportional to the number of scattered particles shared by the two subsystems. The voltage bias and the non-trivial scattering constitute necessary and generically-sufficient conditions for the extensive terms in Eqs. (7)–(10) to not vanish.

To support our general analytical results, we compared them to numerics for a specific model where the scattering is a result of a single impurity at the site $m = 0$.[2] For this model, $m_0 = 0$ and $\mathcal{H}_{\text{scat}} = \epsilon_0 c_0^\dagger c_0$ in Eq. (4), $\epsilon_0$ being the impurity energy. The scattering matrix for this model yields the transmission probability

$$\mathcal{T}(k) = \frac{\sin^2 k}{\sin^2 k + (\epsilon_0/2\eta)^2} . \tag{11}$$

---

[2]The numerical results presented in Figs. 2 and 3 were calculated in the limit $d_i/\ell_i \to \infty$, which allows to simplify the expressions for the elements of two-point correlation matrices, as explained in Appendix A. In Appendix D we also compare these numerical results to those computed for finite $d_i/\ell_i$, demonstrating convergence for $d_i/\ell_i \gg 1$.

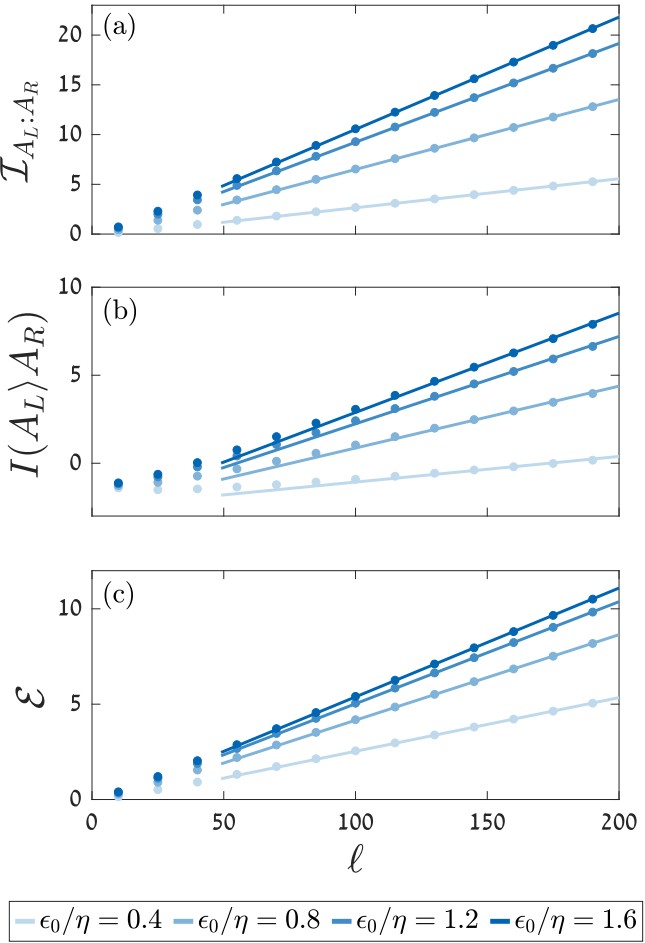

Figure 2: The single impurity model: Scaling of (a) the mutual information, (b) the coherent information, and (c) the fermionic negativity between subsystems $A_L$ and $A_R$ for the symmetric case $\ell_L = \ell_R = \ell$ and $d_L = d_R$, in the limit $d_i \gg \ell_i$. The analytical results of Eqs. (8)–(10) for $\ell \geq 50$ (lines) are compared to numerical results (dots) for different values of the impurity energy $\epsilon_0$, with the Fermi momenta fixed at $k_{F,R} = \pi/2$ and $k_{F,L} = 2\pi/3$.

Good agreement with numerics is manifest in Fig. 2, where, focusing on the aforementioned symmetric case with two intervals of length $\ell$, we plotted the scaling with $\ell$ of all three correlation measures for different ratios of $\epsilon_0/\eta$. The analytical results of Eqs. (8)–(10) are plotted for $\ell \geq 50$ (with a constant-in-$\ell$ additive correction term as the only fitting parameter), as for small values of $\ell$ there is a considerable contribution from subleading terms beyond the leading volume-law term (an exact analytical result for the most dominant subleading term, which is logarithmic in $\ell$, is derived in Ref. [56]).

In Fig. 3 we illustrate a rather counter-intuitive attribute of our results, using the example of the single impurity model. For fixed values of $\ell_L$ and $\ell_R$, we plot the dependence of the MI and the negativity on the positions of the subsystems, and observe that this dependence is non-monotonic. Indeed, Eqs. (8)–(10) state that the long-range correlations are the strongest when the overlap between one subsystem and the mirror image of the other is maximal; if one subsystem is then brought closer to the other, this overlap is reduced and so are the correlations. Fig. 3 again showcases the good agreement of our analytical results with numerics; the apparent slight deviations can be resolved once logarithmic corrections are accounted for [56].

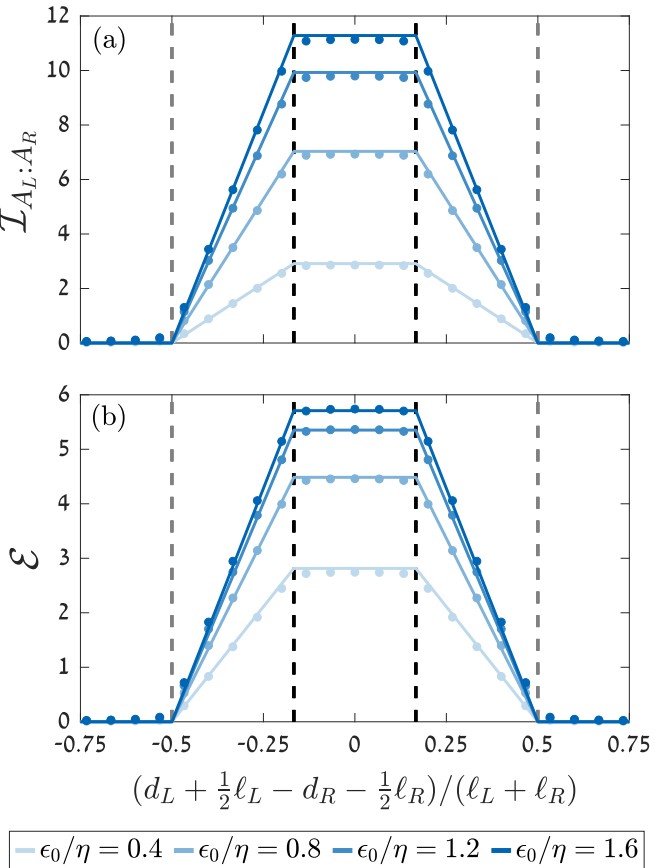

Figure 3: The single impurity model: (a) The mutual information and (b) the fermionic negativity between subsystems $A_L$ and $A_R$ as a function of their positions relative to the impurity. We fix $\ell_L = 100$ and $\ell_R = 200$, and observe the dependence on $d_L - d_R$ in the limit $d_i \gg \ell_i$. Analytical results (lines) are compared to numerical results (dots) for different values of the impurity energy $\epsilon_0$, with the Fermi momenta fixed at $k_{F,R} = \pi/2$ and $k_{F,L} = 2\pi/3$. Letting $\bar{A}_L = \{m | -m \in A_L\}$ denote the mirror image of $A_L$, black dashed vertical lines mark the boundaries of the domain where $\bar{A}_L \subset A_R$, while gray dashed vertical lines mark the boundaries of the domain where $\bar{A}_L \cap A_R \neq \phi$.

## 4 Analytical method

This section delineates the main steps in the derivation of Eqs. (8)–(10), while the discussion of the various technical steps is deferred to the appendices. Subsec. 4.1 focuses on the derivation of the formulae for the MI and CI (both of which are combinations of subsystem entropies), while Subsec. 4.2 deals with the derivation of the negativity asymptotics.

The joint starting point of these computations is the two-point correlation function $\left\langle c_j^\dagger c_m \right\rangle$ for $j, m \in A$. The absence of interactions entails that the states of the total system and its subsystems are Gaussian, and thus entanglement is fully encoded in two-point correlations [43, 44, 57, 58]. The correlation function is given explicitly by

$$\left\langle c_j^\dagger c_m \right\rangle = \int_{-k_{F,R}}^{k_{F,L}} \frac{dk}{2\pi} u_j^*(k) u_m(k) \,, \tag{12}$$

where $u_m(k)$ is the (unnormalized) single-particle wavefunction amplitude at site $m$ of the

scattering state associated with momentum $k$; namely,

$$u_{m>m_0}(k) = \begin{cases} e^{ikm} + r_R(|k|)\,e^{-ikm}, & k < 0, \\ t_L(|k|)\,e^{ikm}, & k > 0, \end{cases}$$

$$u_{m<-m_0}(k) = \begin{cases} t_R(|k|)\,e^{ikm}, & k < 0, \\ e^{ikm} + r_L(|k|)\,e^{-ikm}, & k > 0. \end{cases} \tag{13}$$

Eq. (12) is derived in Appendix A, where we also discuss how this expression may be simplified when $d_i \gg \ell_i$.

## 4.1 Mutual information and coherent information

The exact asymptotics of the MI and of the CI in Eqs. (8) and (10) were obtained through the computation of the Rényi entropies of $A_L$, $A_R$ and $A$. We describe here the main components of this computation, referring the interested reader to Appendix B for full details.

Within the Gaussian steady state, the Rényi entropies of a subsystem $X$ are reduced to functions of $C_X$, the correlation matrix restricted to $X$ ($(C_X)_{jm} = \left\langle c_j^\dagger c_m \right\rangle$ where $j, m \in X$) [57]. Furthermore, these functions admit a simple series expansion, on which our derivation relied. Namely, the Rényi entropies are given by

$$\begin{aligned} S_X^{(n)} &= \frac{1}{1-n}\operatorname{Tr}\ln[(C_X)^n + (\mathbb{I} - C_X)^n] \\ &= \frac{1}{1-n}\sum_{s=1}^{\infty}\frac{(-1)^{s+1}}{s}\operatorname{Tr}\left[\{(C_X)^n + (\mathbb{I} - C_X)^n - \mathbb{I}\}^s\right]. \end{aligned} \tag{14}$$

To obtain an analytical expression for the Rényi entropies, it is therefore sufficient to calculate a general expression for moments $\operatorname{Tr}[(C_X)^p]$, with $p$ being positive integers. Each such moment can be expressed in the form of a $p$-dimensional integral,

$$\operatorname{Tr}\left[(C_X)^p\right] = \int_{[-k_{F,R}, k_{F,L}]^p}\frac{d^p k}{(2\pi)^p}\prod_{j=1}^{p}\left[\sum_{m\in X}u_m(k_{j-1})u_m^*(k_j)\right], \tag{15}$$

where we defined $k_0 = k_p$. Each sum over $m \in X$ in Eq. (15) can be rewritten as an integral over a fictitious variable $\xi_j \in [-1, 1]$, such that $\operatorname{Tr}[(C_X)^p]$ is then expressed as a $2p$-dimensional integral.

The specific form of this integral depends on the choice of $X$, and is relatively involved (see Appendix B). We illustrate schematically the way forward by considering the case of the connected subsystems $X = A_i$. For each of these subsystems, Eq. (15) can be cast in the general form

$$\begin{aligned} \operatorname{Tr}\left[(C_{A_i})^p\right] = \left(\frac{\ell_i}{4\pi}\right)^p \sum_{\vec{\tau}, \vec{\sigma} \in \{0,1\}^{\otimes p}} \int_{[-k_{F,R}, k_{F,L}]^p} d^p k \\ \times \int_{[-1,1]^p} d^p \xi\, f_{\vec{\tau}, \vec{\sigma}}\!\left(\vec{k}\right)\exp\left[i\frac{\ell_i}{2}\sum_{j=1}^{p}\left(k_{\tau_{j-1}} - k_{\sigma_j}\right)\left(\xi_j + 1\right)\right], \end{aligned} \tag{16}$$

where $k_{\sigma_j} = (-1)^{\sigma_j}k_j$, and where the functions $f_{\vec{\tau}, \vec{\sigma}}$ vanish for $\vec{k} \notin [-k_{F,R}, k_{F,L}]^p$ and are independent of $\ell_i$. The origin of the exponential term in Eq. (16) can be traced back to the explicit forms of the wavefunctions $u_m(k)$ in Eq. (13), which are superpositions of $e^{\pm ikm}$.

An integral of the form of Eq. (16) admits a stationary phase approximation in the large-$\ell_i$ limit [29,59,60]; leading-order contributions come only from summands with $\vec{\tau} = \vec{\sigma}$, with the asymptotics of Eq. (16) given by

$$\text{Tr}\big[\big(C_{A_i}\big)^p\big] \sim \ell_i \int_{-k_{F,R}}^{k_{F,L}} \frac{dk_p}{2\pi} \sum_{\vec{\sigma} \in \{0,1\}^{\otimes p}} f_{\vec{\sigma},\vec{\sigma}}\big(k_{\sigma_p}(-1)^{\vec{\sigma}}\big), \tag{17}$$

where $(-1)^{\vec{\sigma}} = ((-1)^{\sigma_1}, \ldots, (-1)^{\sigma_p})$. Substituting the specific expressions for the functions $f_{\vec{\tau},\vec{\sigma}}$ that satisfy Eq. (16), we then find that

$$\text{Tr}\Big[\big\{\big(C_{A_i}\big)^n + \big(\mathbb{I} - C_{A_i}\big)^n - \mathbb{I}\big\}^s\Big] \sim \ell_i \int_{k_-}^{k_+} \frac{dk}{2\pi} \big\{(\mathcal{T}(k))^n + (\mathcal{R}(k))^n - 1\big\}^s, \tag{18}$$

for all positive integers $s$. A similar treatment was applied to compute the leading term in $\Delta \ell_i$ of moments of $C_A$, producing the same result as in Eq. (18) up to replacing $C_{A_i}$ with $C_A$ and $\ell_i$ with $\Delta \ell_L + \Delta \ell_R$.

Summing up these contributions in the series expansion of Eq. (14) yields the extensive terms for the Rényi entropies of $A_L$, $A_R$ and $A$,

$$S_{A_i}^{(n)} \sim \frac{\ell_i}{1-n} \int_{k_-}^{k_+} \frac{dk}{2\pi} \ln\big[(\mathcal{T}(k))^n + (\mathcal{R}(k))^n\big],$$

$$S_A^{(n)} \sim \frac{\Delta \ell_L + \Delta \ell_R}{1-n} \int_{k_-}^{k_+} \frac{dk}{2\pi} \ln\big[(\mathcal{T}(k))^n + (\mathcal{R}(k))^n\big]. \tag{19}$$

The asymptotics in Eq. (19) directly lead to Eq. (7), while Eqs. (8) and (10) are obtained by taking the limit $n \to 1$ and substituting the resulting von Neumann entropies into the definitions in Eqs. (1) and (3). We stress that the universal dependence of the Rényi entropies on the scattering probabilities results from the fact that, at sites $m$ lying outside the scattering region, the wavefunctions $u_m(k)$ in Eq. (13) are written only in terms of plane waves and scattering amplitudes.

## 4.2 Fermionic negativity

Here we outline the principal steps in the derivation of Eq. (9), the asymptotic formula for the fermionic negativity; a more detailed account of the computation appears in Appendix C. The derivation relies on the fact that the negativity $\mathcal{E}$ can be obtained as the analytic continuation of the Rényi negativities $\mathcal{E}_n = \ln \text{Tr}\Big[\big((\tilde{\rho}_A)^\dagger \tilde{\rho}_A\big)^{n/2}\Big]$ at $n = 1$, where $\mathcal{E}_n$ are evaluated at even values of $n$ [43]. In analogy to the Rényi entropies, the Rényi negativities $\mathcal{E}_n$ can be expressed as functions of $C_A$ and of a transformed two-point correlation matrix restricted to $A$ [43,44,58]. As shown in Appendix C, this expression for $\mathcal{E}_n$ leads to the following series expansion:

$$\mathcal{E}_n = \sum_{s=1}^{\infty} \frac{(-1)^{s+1}}{s} \text{Tr}\left[\left\{\prod_{\gamma=-\frac{n-1}{2}}^{\frac{n-1}{2}} \big(\mathbb{I} - C_\gamma\big) - \mathbb{I}\right\}^s\right]. \tag{20}$$

Here each $C_\gamma$ is a transformed version of $C_A$, given by

$$C_\gamma = \begin{pmatrix} \big(1 - e^{\frac{2\pi i \gamma}{n}}\big)\mathbb{I}_{\ell_L} & 0 \\ 0 & \big(1 + e^{\frac{-2\pi i \gamma}{n}}\big)\mathbb{I}_{\ell_R} \end{pmatrix} C_A, \tag{21}$$

where the entries of $C_A$ are ordered such that the first $\ell_L$ indices correspond to sites in $A_L$, and the next $\ell_R$ correspond to sites in $A_R$.

Eq. (20) reduces the calculation of the Rényi negativities to that of terms of the form $\text{Tr}\left[C_{\gamma_1}C_{\gamma_2}\ldots C_{\gamma_p}\right]$, which, by using Eq. (12), may also be written as

$$\text{Tr}\left[C_{\gamma_1}\ldots C_{\gamma_p}\right] = \int \frac{d^p k}{(2\pi)^p} \prod_{j=1}^{p}\left[\left(1-e^{\frac{2\pi i \gamma_j}{n}}\right)\sum_{m\in A_L} u_m(k_{j-1})u_m^*(k_j)\right.$$
$$\left. +\left(1+e^{\frac{-2\pi i \gamma_j}{n}}\right)\sum_{m\in A_R} u_m(k_{j-1})u_m^*(k_j)\right], \qquad (22)$$

where the integral is computed over the domain $\left[-k_{F,R}, k_{F,L}\right]^p$. The remainder of the calculation is similar in spirit to that of the Rényi entropies: the explicit forms of the wavefunctions from Eq. (13) are substituted into Eq. (22); Eq. (22) is rewritten as a $2p$-dimensional integral using $p$ fictitious variables; and finally, this $2p$-dimensional integral is estimated through a stationary phase approximation (see Appendix C). This process eventually leads to the following result for every positive integer $s$:

$$\text{Tr}\left[\left\{\prod_{\gamma=-\frac{n-1}{2}}^{\frac{n-1}{2}}\left(\mathbb{I}-C_\gamma\right)-\mathbb{I}\right\}^s\right] \sim \ell_{\text{mirror}}\int_{k_-}^{k_+}\frac{dk}{2\pi}\left\{\left[\mathcal{T}^{n/2}+\mathcal{R}^{n/2}\right]^2-1\right\}^s$$
$$+(\Delta\ell_L+\Delta\ell_R)\int_{k_-}^{k_+}\frac{dk}{2\pi}\left\{\mathcal{T}^n+\mathcal{R}^n-1\right\}^s. \qquad (23)$$

Upon summation of the series in Eq. (20), we find that the Rényi negativities are given by

$$\mathcal{E}_n \sim \ell_{\text{mirror}}\int_{k_-}^{k_+}\frac{dk}{\pi}\ln\left[\mathcal{T}^{n/2}+\mathcal{R}^{n/2}\right]+(\Delta\ell_L+\Delta\ell_R)\int_{k_-}^{k_+}\frac{dk}{2\pi}\ln\left[\mathcal{T}^n+\mathcal{R}^n\right], \qquad (24)$$

and the exact asymptotics of the negativity in Eq. (9) is obtained once the limit $n\to 1$ is finally taken.

# 5 Discussion and Outlook

In this work we derived the exact asymptotics of correlation measures for a nonequilibrium steady state of 1D noninteracting fermions. We have shown that this state hosts extensive long-range entanglement between subsystems that are on opposite sides of a current-conserving scatterer, provided that their distances from it are similar. The volume-law terms of these measures stem from the extensive number of single-particle wavepackets that originate in the high-chemical-potential reservoir, which are split by the scatterer so that they are in a superposition of being found in either one of the mirroring subsystems. The correlation measures thus exhibit a simple and universal dependence on scattering probabilities, allowing to clearly read off the necessary and sufficient conditions for the generation of this strong long-range entanglement. Apart from the requirement that the scatterer be non-trivial, the essential ingredients are the absence of decoherence mechanisms, along with the extensive excess of particles emerging from one of the reservoirs.

We therefore expect the main features of our results to hold in a wide class of nonequilibrium steady states, including those of integrable interacting systems [8, 10, 61], as well as when the reservoirs are at finite temperatures, and when the scatterer induces particle

gain and loss [29]. Similar features should also appear in the dynamics following a quench where two decoupled half-infinite chains are prepared with unequal fillings, and the scatterer is suddenly introduced [11, 62, 63]. All of these scenarios offer intriguing prospects for future studies. It would also be interesting to study the interplay of the effects uncovered by this work with decoherence, which could arise due to an integrability-breaking impurity or when the system is coupled to Lindblad baths [64–67], or the interplay of the same effects with the addition of quadratic pairing terms [68], which break charge conservation, to the Hamiltonian of Eq. (4).

Realizations of such models with, e.g., ultracold atoms [69] should allow to experimentally extract entanglement measures [70–75]. In this context we highlight our results in Eqs. (19) and (24) for the Rényi versions of these measures, which are generally more amenable to efficient measurement than their von Neumann counterparts.

Replacing the scattering region with a disordered region [30, 33], the signatures of localization and resonances on long-range entanglement properties could also be a fruitful subject of research. Finally, another possible future direction is the study of symmetry-resolution [76–86] of the different entanglement measures analyzed here.

## Acknowledgments

We thank P. Calabrese, V. Eisler, and E. Sela for fruitful discussions.

**Funding information** Our work was supported by the Israel Science Foundation (ISF) and the Directorate for Defense Research and Development (DDR&D) grant No. 3427/21, and by the US-Israel Binational Science Foundation (BSF) Grant No. 2020072. S.F. is supported by the Azrieli Foundation Fellows program.

## A  Two-point correlations

Here we derive Eq. (12), which is the general expression for the two-point correlation function $\left\langle c_j^\dagger c_m \right\rangle$ for sites outside the scattering region, $|j|, |m| > m_0$. We consider a long chain with $N \gg 1$ sites, where the small scattering region is located at its center; in the end we will take the thermodynamic limit $N \to \infty$.

An annihilation operator $c_m$ may be expanded in terms of annihilation operators corresponding to the single-particle energy eigenstates. We include only extended scattering states in this expansion, neglecting the contribution of localized bound states, since the amplitude of a bound state wavefunction at any site outside the scattering region decays exponentially with the distance of that site from the scatterer. More concretely, we associate an annihilation operator $c_{k,L}$ ($c_{k,R}$) with the scattering state of a particle originating in the left (right) reservoir with momentum $k > 0$ ($k < 0$). Then, $c_m$ may be written as follows:

$$c_m = \frac{1}{\sqrt{N}} \left[ \sum_{-\pi < k < 0} u_m(k) c_{k,R} + \sum_{0 < k < \pi} u_m(k) c_{k,L} \right], \tag{A.1}$$

where the wavefunctions $u_m(k)$ are given in Eq. (13). In the language of scattering state creation operators, the nonequilibrium steady state analyzed in this work is given by

$$|\text{NESS}\rangle = \left( \prod_{-k_{F,R} < k < 0} c_{k,R}^\dagger \right) \left( \prod_{0 < k < k_{F,L}} c_{k,L}^\dagger \right) |\text{vac}\rangle, \tag{A.2}$$

with $|\text{vac}\rangle$ being the vacuum state. Substituting Eq. (A.1) into the definition of the two-point correlation function, we find that, in the thermodynamic limit $N \to \infty$, the correlation function approaches the integral expression of Eq. (12).

As explained in Sec. 4, the different correlation measures discussed in this work can all be expressed as functions of two-point correlation matrices restricted to the subsystems of interest. That is, we generally consider the terms given by Eq. (12) only for sites $j, m \in A$, and correlation measures can scale at most as $\mathcal{O}(\ell_L + \ell_R)$, given the dimensions of the correlation matrices. This, in turn, implies that in the limit $d_i/\ell_i \to \infty$ (with $d_L - d_R$ kept fixed), calculations of correlation measures can be simplified by first neglecting certain terms in the expressions for the matrix elements $\left\langle c_j^\dagger c_m \right\rangle$, and only then calculating the appropriate functions of the correlation matrices.

In particular, using Eq. (13) we observe that in Eq. (12) the correlation function is a sum of integrals, where in each integral the integrand is a product of a function of $k$ that is independent of $j, m$ and an exponent of the form $\exp\left[i\alpha_{j,m}k\right]$, with $\alpha_{j,m} \in \{\pm(j \pm m)\}$. Then, the Riemann-Lebesgue lemma leads to the conclusion that when $\left|\alpha_{j,m}\right| \gg \ell_L, \ell_R$, the contribution of the integral is negligible and may be omitted. This entails that when $d_L, d_R \gg \ell_L, \ell_R$ we may use the following approximations for the correlation matrix elements:

$$
\left\langle c_j^\dagger c_m \right\rangle \approx
\begin{cases}
\int_{-k_{F,R}}^{k_{F,R}} \frac{dk}{2\pi} e^{-i(j-m)k} + \int_{k_{F,R}}^{k_{F,L}} \frac{dk}{2\pi} \mathcal{T}(k) e^{-i(j-m)k}, & j, m \in A_R, \\
\int_{-k_{F,L}}^{k_{F,L}} \frac{dk}{2\pi} e^{-i(j-m)k} + \int_{k_{F,L}}^{k_{F,R}} \frac{dk}{2\pi} \mathcal{T}(k) e^{i(j-m)k}, & j, m \in A_L, \\
\int_{k_{F,R}}^{k_{F,L}} \frac{dk}{2\pi} t_L^*(k) r_L(k) e^{-i(j+m)k}, & m \in A_L, \text{ and } j \in A_R, \\
\int_{k_{F,L}}^{k_{F,R}} \frac{dk}{2\pi} t_R^*(k) r_R(k) e^{i(j+m)k}, & j \in A_L, \text{ and } m \in A_R.
\end{cases}
\tag{A.3}
$$

Our analytical results were all derived based on the full expression for $\left\langle c_j^\dagger c_m \right\rangle$ in Eq. (12). As they indicated that the volume-law terms of the different correlation measures depend on $d_L$ and $d_R$ only through $d_L - d_R$, they were compared in Figs. 2–3 to numerical results that were computed in the limit $d_i/\ell_i \to \infty$, based on the approximated correlation function in Eq. (A.3). A comparison to a numerical calculation that relies on the full expression for the correlation function is provided in Appendix D.

## B  Calculation of the Rényi entropies

In this appendix we describe the analytical method used for the computation of the Rényi entropies $S_X^{(n)} = \frac{1}{1-n} \ln \text{Tr}\left[(\rho_X)^n\right]$ for the subsystems $X = A_L, A_R, A$. The final results are given in Eq. (19). The results for the Rényi entropies lead directly to the asymptotics of the MI and CI (Eqs. (8) and (10)), as explained in Subsec. 4.1.

In Subsec. 4.1 we showed that the calculation of Rényi entropies can be reduced to that of the moments $\text{Tr}\left[(C_X)^p\right]$ for all positive integers $p$. We now derive the asymptotics of these moments for the subsystems of interest, starting from their integral expression in Eq. (15). The analysis is based on the stationary phase approximation (SPA) [59], and is inspired by the analytical methods of Refs. [29, 60].

### B.1  Asymptotics of moments for the connected subsystems

We first consider the case $X = A_R$. We begin by introducing the notation

$\mathcal{W}_R(x) = \frac{x}{\sin x} \exp\left[2i\left(m_0 + d_R + \frac{1}{2}\right)x\right]$, and observing that

$$\sum_{m=m_0+d_R+1}^{m_0+d_R+\ell_R} \exp\left[im\left(k_{j-1}-k_j\right)\right] = \frac{\ell_R}{2}\mathcal{W}_R\left(\frac{k_{j-1}-k_j}{2}\right)\int_{-1}^{1} d\xi \exp\left[i\frac{\ell_R}{2}\left(k_{j-1}-k_j\right)(\xi+1)\right]. \quad (B.1)$$

For convenience, we define the notation $k_{a_j} = (-1)^{a_j} k_j$ for $a_j \in \{0,1\}$, as well as $(-1)^{\vec{a}} = ((-1)^{a_1},\ldots,(-1)^{a_p})$ for $\vec{a} \in \{0,1\}^{\otimes p}$. We use Eqs. (13) and (B.1) to write

$$\sum_{m\in A_R} u_m(k_{j-1})u_m^*(k_j) = \frac{\ell_R}{2}\sum_{a_{j-1},b_j=0,1} \Xi^{a_{j-1}b_j}\left(k_{a_{j-1}},k_{b_j}\right)\Theta\left(k_{a_{j-1}}\right)\Theta\left(k_{b_j}\right), \quad (B.2)$$

where $\Theta(x)$ is the Heaviside step function, and where we defined

$$\Xi^{00}\left(k_{j-1},k_j\right) = t_L\left(|k_{j-1}|\right)t_L^*\left(|k_j|\right)\mathcal{W}_R\left(\frac{k_{j-1}-k_j}{2}\right)\int_{-1}^{1} d\xi\, e^{\frac{i}{2}\ell_R(k_{j-1}-k_j)(\xi+1)},$$

$$\Xi^{11}\left(k_{j-1},k_j\right) = \int_{-1}^{1} d\xi\left\{\mathcal{W}_R\left(\frac{k_j-k_{j-1}}{2}\right)e^{\frac{i}{2}\ell_R(k_j-k_{j-1})(\xi+1)}\right.$$

$$\left. + r_R\left(|k_{j-1}|\right)r_R^*\left(|k_j|\right)\mathcal{W}_R\left(\frac{k_{j-1}-k_j}{2}\right)e^{\frac{i}{2}\ell_R(k_{j-1}-k_j)(\xi+1)}\right\}$$

$$+ \int_{-1}^{1} d\xi\left\{r_R^*\left(|k_j|\right)\mathcal{W}_R\left(\frac{-k_{j-1}-k_j}{2}\right)e^{-\frac{i}{2}\ell_R(k_{j-1}+k_j)(\xi+1)}\right.$$

$$\left. + r_R\left(|k_{j-1}|\right)\mathcal{W}_R\left(\frac{k_{j-1}+k_j}{2}\right)e^{\frac{i}{2}\ell_R(k_{j-1}+k_j)(\xi+1)}\right\},$$

$$\Xi^{01}\left(k_{j-1},k_j\right) = \int_{-1}^{1} d\xi\, t_L\left(|k_{j-1}|\right)\left\{\mathcal{W}_R\left(\frac{k_{j-1}+k_j}{2}\right)e^{\frac{i}{2}\ell_R(k_{j-1}+k_j)(\xi+1)}\right.$$

$$\left. + r_R^*\left(|k_j|\right)\mathcal{W}_R\left(\frac{k_{j-1}-k_j}{2}\right)e^{\frac{i}{2}\ell_R(k_{j-1}-k_j)(\xi+1)}\right\}, \quad (B.3)$$

and $\Xi^{10}\left(k_{j-1},k_j\right) = \Xi^{01}\left(k_j,k_{j-1}\right)^*$.

When plugging Eq. (B.2) into the expression for $\text{Tr}\left[\left(C_{A_R}\right)^p\right]$ in Eq. (15), we will generally get a sum of $2p$-dimensional integrals, each of the form

$$\mathcal{F}\left(\vec{\tau},\vec{\sigma}\right) = \left(\frac{\ell_R}{4\pi}\right)^p \int_{[-k_{F,R},k_{F,L}]^p} d^p k \int_{[-1,1]^p} d^p\xi\, f_{\vec{\tau},\vec{\sigma}}\left(\vec{k}\right)\exp\left[i\frac{\ell_R}{2}\sum_{j=1}^{p}\left(k_{\tau_{j-1}}-k_{\sigma_j}\right)(\xi_j+1)\right], \quad (B.4)$$

with $\vec{\tau},\vec{\sigma} \in \{0,1\}^{\otimes p}$, and where the function $f_{\vec{\tau},\vec{\sigma}}\left(\vec{k}\right)$ is independent of $\ell_R$ and supported on $\left[-k_{F,R},k_{F,L}\right]^p$. We apply a change of variables

$$\begin{aligned}\zeta_1 &= \xi_1,\\ \zeta_j &= \xi_j - \xi_{j-1} \quad (2 \le j \le p),\end{aligned} \quad (B.5)$$

and obtain

$$\mathcal{F}\left(\vec{\tau},\vec{\sigma}\right) = \left(\frac{\ell_R}{4\pi}\right)^p \int_{[-k_{F,R},k_{F,L}]^p} d^p k \int d^p\zeta\, f_{\vec{\tau},\vec{\sigma}}\left(\vec{k}\right)$$

$$\times \exp\left[i\frac{\ell_R}{2}\left\{\sum_{j=1}^{p}\left(k_{\tau_{j-1}}-k_{\sigma_j}\right) + \sum_{l=1}^{p}\zeta_l\sum_{j=l}^{p}\left(k_{\tau_{j-1}}-k_{\sigma_j}\right)\right\}\right]. \quad (B.6)$$

These are the integrals to which we apply the SPA. The SPA allows us to detect which integrals contribute to the leading-order terms of $\text{Tr}\big[\big(C_{A_R}\big)^p\big]$, and to compute their exact contribution to the linear term in $\ell_R$. From Eq. (B.6) it is evident that the answer to the question of whether $\mathcal{F}\big(\vec{\tau},\vec{\sigma}\big)$ has a leading-order contribution is determined by the values of $\vec{\tau}$ and $\vec{\sigma}$. We now illustrate the method by focusing on two concrete cases for the choice of $\vec{\tau}$ and $\vec{\sigma}$.

Assuming that $\tau_j = \sigma_j$ for every $j$, we find that

$$\mathcal{F}\big(\vec{\sigma},\vec{\sigma}\big) = \left(\frac{\ell_R}{4\pi}\right)^p \int_{-k_{F,R}}^{k_{F,L}} dk_p \int_{-1}^{1} d\zeta_1 \int d^{p-1}k \, d^{p-1}\zeta \, f_{\vec{\sigma},\vec{\sigma}}\big(\vec{k}\big) \exp\left[i\frac{\ell_R}{2}\sum_{l=2}^{p}\zeta_l\big(k_{\sigma_{l-1}}-k_{\sigma_p}\big)\right]. \tag{B.7}$$

Applying the SPA to the innermost $(2p-2)$-dimensional integral, with respect to the stationary point of the function $\frac{1}{2}\sum_{l=2}^{p}\zeta_l\big(k_{\sigma_{l-1}}-k_{\sigma_p}\big)$ (at which $\zeta_l = 0$ and $k_{\sigma_{l-1}} = k_{\sigma_p}$ for $l = 2,\ldots,p$), we obtain [59]

$$\mathcal{F}\big(\vec{\sigma},\vec{\sigma}\big) \sim \left(\frac{\ell_R}{4\pi}\right)^p \int_{-k_{F,R}}^{k_{F,L}} dk_p \int_{-1}^{1} d\zeta_1 f_{\vec{\sigma},\vec{\sigma}}\big(k_{\sigma_p}(-1)^{\vec{\sigma}}\big)\left\{\left(\frac{2\pi}{\ell_R}\right)^{p-1}|\det H|^{-1/2}\right\}$$

$$= \frac{\ell_R}{2\pi}\int_{-k_{F,R}}^{k_{F,L}} dk_p f_{\vec{\sigma},\vec{\sigma}}\big(k_{\sigma_p}(-1)^{\vec{\sigma}}\big), \tag{B.8}$$

where we used the fact that the Hessian $H$ at the stationary point yields $|\det H| = \left(\frac{1}{2}\right)^{2p-2}$.

If, on the other hand, $\tau_j = \sigma_j$ for every $j \geq 2$ but $\tau_1 \neq \sigma_1$, then

$$\mathcal{F}\big(\vec{\tau},\vec{\sigma}\big) = \left(\frac{\ell_R}{4\pi}\right)^p \int_{-k_{F,R}}^{k_{F,L}} dk_1 \int_{-1}^{1} d\zeta_1 \exp\big[i\ell_R k_{\tau_1}(\zeta_1+1)\big]$$

$$\times \int d^{p-1}k \, d^{p-1}\zeta \, f_{\vec{\tau},\vec{\sigma}}\big(\vec{k}\big)\exp\left[i\frac{\ell_R}{2}\sum_{l=2}^{p}\zeta_l\big(k_{\tau_{l-1}}-k_{\tau_p}\big)\right]. \tag{B.9}$$

Applying the SPA to the innermost integral will again produce a factor proportional to $\ell_R^{-p+1}$, yielding

$$\mathcal{F}\big(\vec{\tau},\vec{\sigma}\big) \sim \frac{\ell_R}{4\pi}\int_{-k_{F,R}}^{k_{F,L}} dk_1 \int_{-1}^{1} d\zeta_1 f_{\vec{\tau},\vec{\sigma}}\big(k_{\tau_1}(-1)^{\vec{\tau}}\big)\exp\big[i\ell_R k_{\tau_1}(\zeta_1+1)\big], \tag{B.10}$$

only now the remaining phase factor $\exp\big[i\ell_R k_{\tau_1}(\zeta_1+1)\big]$ will eliminate the extensive contribution, such that $\mathcal{F}\big(\vec{\tau},\vec{\sigma}\big)$ can have a contribution that is, at most, constant in $\ell_R$.

From these two examples, it is straightforward to infer the more general rule that an integral $\mathcal{F}\big(\vec{\tau},\vec{\sigma}\big)$ can contribute to $\text{Tr}\big[\big(C_{A_R}\big)^p\big]$ beyond the constant-in-$\ell_R$ order only if $\vec{\sigma} = \vec{\tau}$. Furthermore, if indeed $\vec{\sigma} = \vec{\tau}$, Eq. (B.8) indicates the contribution of $\mathcal{F}\big(\vec{\tau},\vec{\sigma}\big)$ to the linear-in-$\ell_R$ term of $\text{Tr}\big[\big(C_{A_R}\big)^p\big]$.

Let us now apply this general conclusion to our problem. Substituting Eq. (B.2) into Eq. (15), we obtain

$$\text{Tr}\big[\big(C_{A_R}\big)^p\big] = \left(\frac{\ell_R}{4\pi}\right)^p \int_{[-k_{F,R},k_{F,L}]^p} d^p k \sum_{\vec{a}\in\{0,1\}^{\otimes p}} \prod_{j=1}^{p}\Big[\Xi^{a_{j-1}a_j}\big(k_{a_{j-1}},k_{a_j}\big)\Theta\big(k_{a_j}\big)\Big]. \tag{B.11}$$

Eq. (B.8) tells us that the focus on leading-order terms confines $\mathcal{F}\big(\vec{\sigma},\vec{\sigma}\big)$ to an integration subdomain where $k_{\sigma_j} = k_{\sigma_p}$ for all $1 \leq j \leq p$; this implies that, for the purpose of calculating

the leading-order asymptotics of $\text{Tr}\big[\big(C_{A_R}\big)^p\big]$, some of the terms in the full expressions for $\Xi^{a_{j-1}a_j}\big(k_{a_{j-1}}, k_{a_j}\big)$ in Eq. (B.3) may be *a priori* discarded, given that for them the $k_{\sigma_{j-1}} = k_{\sigma_j}$ requirement is satisfied only when $k_{\sigma_{j-1}} = k_{\sigma_j} = 0$. Namely, we may replace

$$
\Xi^{11}\big(k_{j-1}, k_j\big) \longrightarrow \int_{-1}^{1} d\xi \bigg\{ \mathcal{W}_R\bigg(\frac{k_j - k_{j-1}}{2}\bigg) e^{\frac{i}{2}\ell_R(k_j - k_{j-1})(\xi+1)}
$$
$$
+ r_R\big(|k_{j-1}|\big) r_R^*\big(|k_j|\big) \mathcal{W}_R\bigg(\frac{k_{j-1} - k_j}{2}\bigg) e^{\frac{i}{2}\ell_R(k_{j-1} - k_j)(\xi+1)} \bigg\},
$$
$$
\Xi^{01}\big(k_{j-1}, k_j\big) \longrightarrow \int_{-1}^{1} d\xi \, t_L\big(|k_{j-1}|\big) r_R^*\big(|k_j|\big) \mathcal{W}_R\bigg(\frac{k_{j-1} - k_j}{2}\bigg) e^{\frac{i}{2}\ell_R(k_{j-1} - k_j)(\xi+1)}, \quad \text{(B.12)}
$$

and again $\Xi^{10}\big(k_{j-1}, k_j\big) = \Xi^{01}\big(k_j, k_{j-1}\big)^*$. Note that in the first summand appearing in the expression for $\Xi^{11}\big(k_{j-1}, k_j\big)$, the term in the exponent has an opposite sign compared to all other integrals (including $\Xi^{00}\big(k_{j-1}, k_j\big)$). Since we have established that expressions of the form $\mathcal{F}\big(\vec{\tau}, \vec{\sigma}\big)$ in Eq. (B.4) contribute to the leading order only when $\vec{\sigma} = \vec{\tau}$, a leading-order contribution to Eq. (B.11) will arise from this integral only for $\vec{a} = 1^{\otimes p}$, meaning that we can write

$$
\text{Tr}\big[\big(C_{A_R}\big)^p\big] \sim \bigg(\frac{\ell_R}{4\pi}\bigg)^p \int_{[-k_{F,R}, k_{F,L}]^p} d^p k \int_{[-1,1]^p} d^p \xi
$$
$$
\times \prod_{j=1}^{p} \bigg[ \Theta(-k_j) \mathcal{W}_R\bigg(\frac{k_{j-1} - k_j}{2}\bigg) \exp\bigg[\frac{i\ell_R}{2}\big(k_{j-1} - k_j\big)\big(\xi_j + 1\big)\bigg] \bigg]
$$
$$
+ \bigg(\frac{\ell_R}{4\pi}\bigg)^p \int_{[-k_{F,R}, k_{F,L}]^p} d^p k \int_{[-1,1]^p} d^p \xi \sum_{\vec{a} \in \{0,1\}^{\otimes p}} \prod_{j=1}^{p} \bigg[ \Theta\big(k_{a_j}\big) \mathcal{W}_R\bigg(\frac{k_{a_{j-1}} - k_{a_j}}{2}\bigg)
$$
$$
\times \exp\bigg[\frac{i\ell_R}{2}\big(k_{a_{j-1}} - k_{a_j}\big)\big(\xi_j + 1\big)\bigg] \bigg] \frac{1 + (-1)^{a_j}\big[\mathcal{T}\big(k_{a_j}\big) - \mathcal{R}\big(k_{a_j}\big)\big]}{2}. \quad \text{(B.13)}
$$

Applying the SPA as explained above while using the fact that $\mathcal{W}_R(0) = 1$, we thus have

$$
\text{Tr}\big[\big(C_{A_R}\big)^p\big] \sim \ell_R \begin{cases} \frac{k_{F,R}}{\pi} + \int_{k_{F,R}}^{k_{F,L}} \frac{dk}{2\pi} \big(\mathcal{T}(k)\big)^p, & k_{F,L} > k_{F,R}, \\ \frac{k_{F,L} + k_{F,R}}{2\pi} + \int_{k_{F,L}}^{k_{F,R}} \frac{dk}{2\pi} \big(\mathcal{R}(k)\big)^p, & k_{F,L} < k_{F,R}. \end{cases} \quad \text{(B.14)}
$$

The derivation for the case $X = A_L$ is equivalent, yielding

$$
\text{Tr}\big[\big(C_{A_L}\big)^p\big] \sim \ell_L \begin{cases} \frac{k_{F,L} + k_{F,R}}{2\pi} + \int_{k_{F,R}}^{k_{F,L}} \frac{dk}{2\pi} \big(\mathcal{R}(k)\big)^p, & k_{F,L} > k_{F,R}, \\ \frac{k_{F,L}}{\pi} + \int_{k_{F,L}}^{k_{F,R}} \frac{dk}{2\pi} \big(\mathcal{T}(k)\big)^p, & k_{F,L} < k_{F,R}. \end{cases} \quad \text{(B.15)}
$$

## B.2 Asymptotics of moments for the disjoint subsystem

We now consider the case $X = A$. In the summation over sites $m$ in $\sum_{m \in A} u_m\big(k_{j-1}\big) u_m^*\big(k_j\big)$ (the sum that appears in Eq. (15)) we will separate mirroring sites from sites which are not mirrored. For concreteness, we assume that $d_L < d_R < d_L + \ell_L < d_R + \ell_R$, where the subsequent

generalization is straightforward. We then have

$$\sum_{m \in A} u_m(k_{j-1}) u_m^*(k_j) = \sum_{m=m_0+d_L+1}^{m_0+d_R} u_{-m}(k_{j-1}) u_{-m}^*(k_j) + \sum_{m=m_0+d_L+\ell_L+1}^{m_0+d_R+\ell_R} u_m(k_{j-1}) u_m^*(k_j)$$

$$+ \sum_{m=m_0+d_R+1}^{m_0+d_L+\ell_L} \left[ u_m(k_{j-1}) u_m^*(k_j) + u_{-m}(k_{j-1}) u_{-m}^*(k_j) \right]. \qquad \text{(B.16)}$$

We define the function $\mathcal{W}_L(x) = \frac{x}{\sin x} \exp\left[2i\left(m_0+d_L+\frac{1}{2}\right)x\right]$. Sums of exponents appearing in Eq. (B.16) can be written as integrals:

$$\sum_{m=m_0+d_L+1}^{m_0+d_R} \exp\left[im(k_{j-1}-k_j)\right] = \frac{\Delta \ell_L}{2} \mathcal{W}_L\left(\frac{k_{j-1}-k_j}{2}\right) \int_{-1}^{1} d\xi \exp\left\{i(k_{j-1}-k_j)\left[\frac{\Delta \ell_L}{2}(\xi+1)\right]\right\},$$

$$\sum_{m=m_0+d_R+1}^{m_0+d_L+\ell_L} \exp\left[im(k_{j-1}-k_j)\right] = \frac{\ell_{\text{mirror}}}{2} \mathcal{W}_L\left(\frac{k_{j-1}-k_j}{2}\right) \int_{-1}^{1} d\xi \exp\left\{i(k_{j-1}-k_j)\left[\frac{\ell_{\text{mirror}}}{2}(\xi+1)+\Delta \ell_L\right]\right\},$$

$$\sum_{m=m_0+d_L+\ell_L+1}^{m_0+d_R+\ell_R} \exp\left[im(k_{j-1}-k_j)\right] = \frac{\Delta \ell_R}{2} \mathcal{W}_L\left(\frac{k_{j-1}-k_j}{2}\right) \int_{-1}^{1} d\xi \exp\left\{i(k_{j-1}-k_j)\left[\frac{\Delta \ell_R}{2}(\xi+1)+\ell_L\right]\right\}.$$

$$\text{(B.17)}$$

The substitution of Eq. (B.16) into the integral expression for $\text{Tr}[(C_A)^p]$ in Eq. (15) will then yield a sum of integrals of the form

$$\mathcal{F}\left(\overrightarrow{\tau}, \overrightarrow{\sigma}, \overrightarrow{\mathcal{A}}\right) = \left[\prod_{j=1}^{p} \frac{\mathcal{A}_j}{2}\right] \int_{[-k_{F,R}, k_{F,L}]^p} \frac{d^p k}{(2\pi)^p} \int_{[-1,1]^p} d^p \xi \, f\left(\overrightarrow{k}\right)$$

$$\times \exp\left\{i \sum_{j=1}^{p} \left(k_{\tau_{j-1}} - k_{\sigma_j}\right)\left[\frac{\mathcal{A}_j}{2}(\xi_j+1) + \mathcal{B}_j\right]\right\}, \qquad \text{(B.18)}$$

where $(\mathcal{A}_j, \mathcal{B}_j) \in \{(\Delta\ell_L, 0), (\ell_{\text{mirror}}, \Delta\ell_L), (\Delta\ell_R, \ell_L)\}$. Writing $\mathcal{A}_j = \alpha_j \ell$ with $\alpha_j$ being some fixed ratios, we are interested in the leading-order behavior as $\ell \to \infty$. Again defining the variables $\{\zeta_j\}$ as in Eq. (B.5), we arrive at the crucial observation that unless $\mathcal{A}_1 = \mathcal{A}_2 = \ldots = \mathcal{A}_p$ (and hence also $\mathcal{B}_1 = \mathcal{B}_2 = \ldots = \mathcal{B}_p$), we cannot find in the exponent a $(2p-2)$-variable function with a stationary point as before, regardless of the values of $\overrightarrow{\tau}, \overrightarrow{\sigma}$. Leading-order contributions will therefore arise only from terms where $\mathcal{A}_1 = \mathcal{A}_2 = \ldots = \mathcal{A}_p$. We can thus conclude that

$$\text{Tr}[(C_A)^p] \sim \int_{[-k_{F,R}, k_{F,L}]^p} \frac{d^p k}{(2\pi)^p} \prod_{j=1}^{p} \left\{ \sum_{m=m_0+d_L+1}^{m_0+d_R} u_{-m}(k_{j-1}) u_{-m}^*(k_j) \right\}$$

$$+ \int_{[-k_{F,R}, k_{F,L}]^p} \frac{d^p k}{(2\pi)^p} \prod_{j=1}^{p} \left\{ \sum_{m=m_0+d_L+\ell_L+1}^{m_0+d_R+\ell_R} u_m(k_{j-1}) u_m^*(k_j) \right\} + \mathcal{M}^{(p)}, \qquad \text{(B.19)}$$

where we defined

$$\mathcal{M}^{(p)} = \int_{[-k_{F,R}, k_{F,L}]^p} \frac{d^p k}{(2\pi)^p} \prod_{j=1}^{p} \left\{ \sum_{m=m_0+d_R+1}^{m_0+d_L+\ell_L} \left[ u_m(k_{j-1}) u_m^*(k_j) + u_{-m}(k_{j-1}) u_{-m}^*(k_j) \right] \right\}.$$

$$\text{(B.20)}$$

The first two integrals in Eq. (B.19) can be treated in the same way in which the equivalent integrals were treated in the cases of the connected subsystems. What therefore remains to be done is to treat $\mathcal{M}^{(p)}$. We define $\widetilde{\mathcal{W}}(x) = \mathcal{W}_L(x) e^{2ix\Delta\ell_L}$ in order to simplify the notation. In analogy to Eqs. (B.11) and (B.12), we may discard terms that have no leading-order contribution to $\mathcal{M}^{(p)}$ and write

$$\mathcal{M}^{(p)} \sim \left(\frac{\ell_{\text{mirror}}}{4\pi}\right)^p \int_{[-k_{F,R},k_{F,L}]^p} d^p k \sum_{\vec{a}\in\{0,1\}^{\otimes p}} \prod_{j=1}^{p} \left[\widetilde{\Xi}^{a_{j-1}a_j}\left(k_{a_{j-1}},k_{a_j}\right)\Theta\left(k_{a_j}\right)\right], \qquad \text{(B.21)}$$

where

$$\widetilde{\Xi}^{00}\left(k_{j-1},k_j\right) = \left[t_L\left(|k_{j-1}|\right)t_L^*\left(|k_j|\right) + r_L\left(|k_{j-1}|\right)r_L^*\left(|k_j|\right)\right]\widetilde{\mathcal{W}}\left(\frac{k_{j-1}-k_j}{2}\right)$$

$$\times \int_{-1}^{1} d\xi\, e^{\frac{i}{2}\ell_{\text{mirror}}(k_{j-1}-k_j)(\xi+1)} + \widetilde{\mathcal{W}}\left(\frac{k_j-k_{j-1}}{2}\right)\int_{-1}^{1} d\xi\, e^{\frac{i}{2}\ell_{\text{mirror}}(k_j-k_{j-1})(\xi+1)},$$

$$\widetilde{\Xi}^{11}\left(k_{j-1},k_j\right) = \left[t_R\left(|k_{j-1}|\right)t_R^*\left(|k_j|\right) + r_R\left(|k_{j-1}|\right)r_R^*\left(|k_j|\right)\right]\widetilde{\mathcal{W}}\left(\frac{k_{j-1}-k_j}{2}\right)$$

$$\times \int_{-1}^{1} d\xi\, e^{\frac{i}{2}\ell_{\text{mirror}}(k_{j-1}-k_j)(\xi+1)} + \widetilde{\mathcal{W}}\left(\frac{k_j-k_{j-1}}{2}\right)\int_{-1}^{1} d\xi\, e^{\frac{i}{2}\ell_{\text{mirror}}(k_j-k_{j-1})(\xi+1)},$$

$$\widetilde{\Xi}^{01}\left(k_{j-1},k_j\right) = \left[t_L\left(|k_{j-1}|\right)r_R^*\left(|k_j|\right) + r_L\left(|k_{j-1}|\right)t_R^*\left(|k_j|\right)\right]\widetilde{\mathcal{W}}\left(\frac{k_{j-1}-k_j}{2}\right)$$

$$\times \int_{-1}^{1} d\xi\, e^{\frac{i}{2}\ell_{\text{mirror}}(k_{j-1}-k_j)(\xi+1)}, \qquad \text{(B.22)}$$

and $\widetilde{\Xi}^{10}\left(k_{j-1},k_j\right) = \widetilde{\Xi}^{01}\left(k_j,k_{j-1}\right)^*$. Applying the SPA through the same procedure as before, while recalling the unitarity of the scattering matrix in Eq. (5), we then obtain

$$\mathcal{M}^{(p)} \sim \frac{k_{F,L}+k_{F,R}}{\pi}\ell_{\text{mirror}}, \qquad \text{(B.23)}$$

so that in total,

$$\text{Tr}[(C_A)^p]$$

$$\sim \begin{cases} \frac{k_{F,L}+k_{F,R}}{2\pi}(\ell_L+\ell_R) + \Delta\ell_L \int_{k_{F,R}}^{k_{F,L}} \frac{dk}{2\pi}(\mathcal{R}(k))^p + \Delta\ell_R \int_{k_{F,R}}^{k_{F,L}} \frac{dk}{2\pi}\left[(\mathcal{T}(k))^p - 1\right], & k_{F,L} > k_{F,R}, \\ \frac{k_{F,L}+k_{F,R}}{2\pi}(\ell_L+\ell_R) + \Delta\ell_L \int_{k_{F,L}}^{k_{F,R}} \frac{dk}{2\pi}\left[(\mathcal{T}(k))^p - 1\right] + \Delta\ell_R \int_{k_{F,L}}^{k_{F,R}} \frac{dk}{2\pi}(\mathcal{R}(k))^p, & k_{F,L} < k_{F,R}. \end{cases}$$
$$\text{(B.24)}$$

## B.3  Asymptotics of the Rényi entropies

Finally, we use the asymptotics of the moments in Eqs. (B.14), (B.15) and (B.24) to derive the Rényi entropies of the subsystems of interest. In particular, we observe that the terms comprising the series expansion in Eq. (14) follow the asymptotic scaling

$$\text{Tr}\left[\left\{(C_{A_i})^n + (\mathbb{I}-C_{A_i})^n - \mathbb{I}\right\}^s\right] \sim \ell_i \int_{k_-}^{k_+} \frac{dk}{2\pi}\left\{(\mathcal{T}(k))^n + (\mathcal{R}(k))^n - 1\right\}^s,$$

$$\text{Tr}\left[\left\{(C_A)^n + (\mathbb{I}-C_A)^n - \mathbb{I}\right\}^s\right] \sim (\Delta\ell_L + \Delta\ell_R)\int_{k_-}^{k_+} \frac{dk}{2\pi}\left\{(\mathcal{T}(k))^n + (\mathcal{R}(k))^n - 1\right\}^s. \qquad \text{(B.25)}$$

This yields Eq. (19), which is true for any $d_L$ and $d_R$.

# C Calculation of the fermionic negativity

In this appendix we summarize the derivation of the result for the fermionic negativity $\mathcal{E}$ between $A_L$ and $A_R$. The analytical method we employed is similar to that used in Appendix B for the calculation of the Rényi entropies, as we explain below.

The fermionic negativity $\mathcal{E}$ can be obtained from the Rényi negativities

$$\mathcal{E}_n = \ln \mathrm{Tr} \left[ \left( (\tilde{\rho}_A)^\dagger \, \tilde{\rho}_A \right)^{n/2} \right], \tag{C.1}$$

by evaluating $\mathcal{E}_n$ at even values of $n$ and performing an analytic continuation to $n = 1$. $\mathcal{E}_n$ can be written in terms of the restricted correlation matrix $C_A$ and a transformed correlation matrix $C_\Xi$, facilitating a significant simplification of the calculation as in the case of the Rényi entropies. We write

$$C_A = \begin{pmatrix} C_{A_L} & C_{LR} \\ C_{RL} & C_{A_R} \end{pmatrix}, \tag{C.2}$$

where the matrices $C_{LR}$ and $C_{RL} = (C_{LR})^\dagger$ represent two-point correlations between a site in $A_L$ and another in $A_R$, and we define

$$C_\Xi = \frac{1}{2} \left[ \mathbb{I} - (\mathbb{I} + \Gamma_+ \Gamma_-)^{-1} (\Gamma_+ + \Gamma_-) \right], \tag{C.3}$$

where

$$\Gamma_\pm = \begin{pmatrix} 2C_{A_L} - \mathbb{I}_{\ell_L} & \mp 2i C_{LR} \\ \mp 2i C_{RL} & \mathbb{I}_{\ell_R} - 2C_{A_R} \end{pmatrix}. \tag{C.4}$$

The Rényi negativities can then be written as [43, 44, 58]

$$\mathcal{E}_n = \ln \det \left[ (C_\Xi)^{n/2} + (\mathbb{I} - C_\Xi)^{n/2} \right] + \frac{n}{2} \ln \det \left[ (C_A)^2 + (\mathbb{I} - C_A)^2 \right]. \tag{C.5}$$

We now define the polynomials

$$p_n(z) = z^n + (1-z)^n = \prod_{\gamma=-\frac{n-1}{2}}^{\frac{n-1}{2}} \left( 1 - \frac{z}{z_\gamma} \right), \tag{C.6}$$

and

$$\tilde{p}_n(z) = z^{n/2} + (1-z)^{n/2} = \prod_{\gamma=1/2}^{\frac{n-1}{2}} \left( 1 - \frac{z}{\tilde{z}_\gamma} \right), \tag{C.7}$$

for any even integer $n$. Here $\{z_\gamma\}$ and $\{\tilde{z}_\gamma\}$ are, respectively, the roots of $p_n$ and $\tilde{p}_n$, and they satisfy

$$\begin{aligned} (z_\gamma)^{-1} &= 1 - e^{2\pi i \gamma/n}, & \gamma &= -\frac{n-1}{2}, -\frac{n-3}{2}, \ldots, \frac{n-1}{2}, \\ (\tilde{z}_\gamma)^{-1} &= \frac{e^{2\pi i \gamma/n} + e^{-2\pi i \gamma/n}}{e^{2\pi i \gamma/n}}, & \gamma &= \frac{1}{2}, \frac{3}{2}, \ldots, \frac{n-1}{2}. \end{aligned} \tag{C.8}$$

Note that $p_n$ has $n$ different roots, while $\tilde{p}_n$ has $n/2$ roots if $n = 0 \bmod 4$, and $n/2 - 1$ roots if $n = 2 \bmod 4$; in the latter case the missing root corresponds to the index $\gamma = n/4$, for which $1/\tilde{z}_\gamma = 0$. Additionally, we recognize that $\det[\mathbb{I} + \Gamma_+ \Gamma_-] = \det[\mathbb{I} + (\mathbb{I} - 2C_A)^2]$, so that using the definition of $\tilde{p}_n$ we may write the Rényi negativities in Eq. (C.5) as

$$\mathcal{E}_n = \ln \det \left[ \prod_{\gamma=1/2}^{\frac{n-1}{2}} \left[ \frac{\mathbb{I} + \Gamma_+ \Gamma_-}{2} - \frac{(\mathbb{I} - \Gamma_+)(\mathbb{I} - \Gamma_-)}{4\tilde{z}_\gamma} \right] \right]. \tag{C.9}$$

Now, if we define the modified correlation matrices

$$C_\gamma = \begin{pmatrix} \left(1-e^{\frac{2\pi i \gamma}{n}}\right)\mathbb{I}_{\ell_L} & 0 \\ 0 & \left(1+e^{\frac{-2\pi i \gamma}{n}}\right)\mathbb{I}_{\ell_R} \end{pmatrix} C_A = \begin{pmatrix} \left(1-e^{\frac{2\pi i \gamma}{n}}\right)C_{A_L} & \left(1-e^{\frac{2\pi i \gamma}{n}}\right)C_{LR} \\ \left(1+e^{\frac{-2\pi i \gamma}{n}}\right)C_{RL} & \left(1+e^{\frac{-2\pi i \gamma}{n}}\right)C_{A_R} \end{pmatrix},$$

$$C_\gamma' = C_A \begin{pmatrix} \left(1-e^{\frac{2\pi i \gamma}{n}}\right)\mathbb{I}_{\ell_L} & 0 \\ 0 & \left(1+e^{\frac{-2\pi i \gamma}{n}}\right)\mathbb{I}_{\ell_R} \end{pmatrix} = \begin{pmatrix} \left(1-e^{\frac{2\pi i \gamma}{n}}\right)C_{A_L} & \left(1+e^{\frac{-2\pi i \gamma}{n}}\right)C_{LR} \\ \left(1-e^{\frac{2\pi i \gamma}{n}}\right)C_{RL} & \left(1+e^{\frac{-2\pi i \gamma}{n}}\right)C_{A_R} \end{pmatrix},$$

(C.10)

one may check that

$$\frac{\mathbb{I}+\Gamma_+\Gamma_-}{2} - \frac{(\mathbb{I}-\Gamma_+)(\mathbb{I}-\Gamma_-)}{4\tilde{z}_\gamma} = \begin{pmatrix} i\mathbb{I}_{\ell_L} & 0 \\ 0 & \mathbb{I}_{\ell_R} \end{pmatrix}\left(\mathbb{I}-C_\gamma'\right)\begin{pmatrix} -e^{\frac{-4\pi i \gamma}{n}}\mathbb{I}_{\ell_L} & 0 \\ 0 & \mathbb{I}_{\ell_R} \end{pmatrix}\left(\mathbb{I}-C_{\gamma-\frac{n}{2}}\right)\begin{pmatrix} -i\mathbb{I}_{\ell_L} & 0 \\ 0 & \mathbb{I}_{\ell_R} \end{pmatrix}.$$

(C.11)

By substituting Eq. (C.11) into Eq. (C.9), and recognizing that $\prod_{\gamma=1/2}^{(n-1)/2}\left(-e^{\frac{-4\pi i \gamma}{n}}\right)=1$, we arrive at the result

$$\mathcal{E}_n = \ln\det\left[\prod_{\gamma=1/2}^{\frac{n-1}{2}}\left(\mathbb{I}-C_\gamma'\right)\left(\mathbb{I}-C_{-\gamma}\right)\right] = \mathrm{Tr}\ln\left[\prod_{\gamma=-\frac{n-1}{2}}^{\frac{n-1}{2}}\left(\mathbb{I}-C_\gamma\right)\right],$$

(C.12)

which then yields the series expansion of $\mathcal{E}_n$ reported in Eq. (20). As explained in Subsec. 4.2, writing this series expansion reduces the calculation to that of terms of the form $\mathrm{Tr}\left[C_{\gamma_1}C_{\gamma_2}\ldots C_{\gamma_p}\right]$, corresponding to the general integral expression in Eq. (22).

We proceed by applying the SPA to the integrals appearing in Eq. (22). For concreteness we again assume that $d_L < d_R < d_L + \ell_L < d_R + \ell_R$. The same argument that led to Eq. (B.19) allows us to separate the integral into independent leading-order contributions arising from mirrored and unmirrored sites. Namely, to the linear order in $\Delta\ell_L$, $\Delta\ell_R$ and $\ell_{\mathrm{mirror}}$, we have

$$\mathrm{Tr}\left[C_{\gamma_1}\ldots C_{\gamma_p}\right] \sim \int_{[-k_{F,R},k_{F,L}]^p}\frac{d^p k}{(2\pi)^p}\prod_{j=1}^{p}\left[\left(1-e^{\frac{2\pi i \gamma_j}{n}}\right)\sum_{m=m_0+d_L+1}^{m_0+d_R}u_{-m}(k_{j-1})u_{-m}^*(k_j)\right]$$

$$+ \int_{[-k_{F,R},k_{F,L}]^p}\frac{d^p k}{(2\pi)^p}\prod_{j=1}^{p}\left[\left(1+e^{\frac{-2\pi i \gamma_j}{n}}\right)\sum_{m=m_0+d_L+\ell_L+1}^{m_0+d_R+\ell_R}u_{m}(k_{j-1})u_{m}^*(k_j)\right]$$

$$+ \mathcal{M}_{\gamma_1\ldots\gamma_p},$$

(C.13)

where we defined

$$\mathcal{M}_{\gamma_1\ldots\gamma_p} = \int_{[-k_{F,R},k_{F,L}]^p}\frac{d^p k}{(2\pi)^p}\prod_{j=1}^{p}\left\{\sum_{m=m_0+d_R+1}^{m_0+d_L+\ell_L}\left[\left(1-e^{\frac{2\pi i \gamma_j}{n}}\right)u_{-m}(k_{j-1})u_{-m}^*(k_j)\right.\right.$$

$$\left.\left. + \left(1+e^{\frac{-2\pi i \gamma_j}{n}}\right)u_{m}(k_{j-1})u_{m}^*(k_j)\right]\right\}.$$

(C.14)

The first two integrals in Eq. (C.13) have already been treated using the SPA in Appendix B, as they arise (up to a multiplicative constant) in the calculation of the Rényi entropies (see Eqs. (B.14) and (B.15)). Assuming for concreteness that $k_{F,L} > k_{F,R}$, we may therefore write

$$\mathrm{Tr}\left[C_{\gamma_1}\ldots C_{\gamma_p}\right] \sim \Delta\ell_L\left[\frac{k_{F,R}+k_{F,L}}{2\pi} + \int_{k_{F,R}}^{k_{F,L}}\frac{dk}{2\pi}(\mathcal{R}(k))^p\right]\prod_{j=1}^{p}\left(1-e^{\frac{2\pi i \gamma_j}{n}}\right)$$

$$+ \Delta\ell_R\left[\frac{k_{F,R}}{\pi} + \int_{k_{F,R}}^{k_{F,L}}\frac{dk}{2\pi}(\mathcal{T}(k))^p\right]\prod_{j=1}^{p}\left(1+e^{\frac{-2\pi i \gamma_j}{n}}\right) + \mathcal{M}_{\gamma_1\ldots\gamma_p}.$$

(C.15)

Let us now address the asymptotics of $\mathcal{M}_{\gamma_1\dots\gamma_p}$, to the linear order in $\ell_{\mathrm{mirror}}$. Repeating the argument in Appendix B leading to Eq. (B.12), which was also used to obtain Eq. (B.21), we use the SPA to discard terms with no leading-order contribution and write

$$\mathcal{M}_{\gamma_1\dots\gamma_p} \sim \left(\frac{\ell_{\mathrm{mirror}}}{4\pi}\right)^p \int_{[-k_{F,R},k_{F,L}]^p} d^p k \sum_{\vec{a}\in\{0,1\}^{\otimes p}} \prod_{j=1}^p \left[\widetilde{\Xi}_{\gamma_j}^{a_{j-1}a_j}\left(k_{a_{j-1}},k_{a_j}\right)\Theta\left(k_{a_j}\right)\right], \qquad \text{(C.16)}$$

with

$$\widetilde{\Xi}_\gamma^{00}(k_{j-1},k_j) = \left[\left(1+e^{\frac{-2\pi i\gamma}{n}}\right)t_L\left(|k_{j-1}|\right)t_L^*\left(|k_j|\right)+\left(1-e^{\frac{2\pi i\gamma}{n}}\right)r_L\left(|k_{j-1}|\right)r_L^*\left(|k_j|\right)\right]$$
$$\times \widetilde{\mathcal{W}}\left(\frac{k_{j-1}-k_j}{2}\right)\int_{-1}^1 d\xi\, e^{\frac{i}{2}\ell_{\mathrm{mirror}}(k_{j-1}-k_j)(\xi+1)}$$
$$+\left(1-e^{\frac{2\pi i\gamma}{n}}\right)\widetilde{\mathcal{W}}\left(\frac{k_j-k_{j-1}}{2}\right)\int_{-1}^1 d\xi\, e^{\frac{i}{2}\ell_{\mathrm{mirror}}(k_j-k_{j-1})(\xi+1)},$$

$$\widetilde{\Xi}_\gamma^{11}(k_{j-1},k_j) = \left[\left(1-e^{\frac{2\pi i\gamma}{n}}\right)t_R\left(|k_{j-1}|\right)t_R^*\left(|k_j|\right)+\left(1+e^{\frac{-2\pi i\gamma}{n}}\right)r_R\left(|k_{j-1}|\right)r_R^*\left(|k_j|\right)\right]$$
$$\times \widetilde{\mathcal{W}}\left(\frac{k_{j-1}-k_j}{2}\right)\int_{-1}^1 d\xi\, e^{\frac{i}{2}\ell_{\mathrm{mirror}}(k_{j-1}-k_j)(\xi+1)}$$
$$+\left(1+e^{\frac{-2\pi i\gamma}{n}}\right)\widetilde{\mathcal{W}}\left(\frac{k_j-k_{j-1}}{2}\right)\int_{-1}^1 d\xi\, e^{\frac{i}{2}\ell_{\mathrm{mirror}}(k_j-k_{j-1})(\xi+1)},$$

$$\widetilde{\Xi}_\gamma^{01}(k_{j-1},k_j) = \left[\left(1+e^{\frac{-2\pi i\gamma}{n}}\right)t_L\left(|k_{j-1}|\right)r_R^*\left(|k_j|\right)+\left(1-e^{\frac{2\pi i\gamma}{n}}\right)r_L\left(|k_{j-1}|\right)t_R^*\left(|k_j|\right)\right]$$
$$\times \widetilde{\mathcal{W}}\left(\frac{k_{j-1}-k_j}{2}\right)\int_{-1}^1 d\xi\, e^{\frac{i}{2}\ell_{\mathrm{mirror}}(k_{j-1}-k_j)(\xi+1)},$$

$$\widetilde{\Xi}_\gamma^{10}(k_{j-1},k_j) = \left[\left(1+e^{\frac{-2\pi i\gamma}{n}}\right)r_R\left(|k_{j-1}|\right)t_L^*\left(|k_j|\right)+\left(1-e^{\frac{2\pi i\gamma}{n}}\right)t_R\left(|k_{j-1}|\right)r_L^*\left(|k_j|\right)\right]$$
$$\times \widetilde{\mathcal{W}}\left(\frac{k_{j-1}-k_j}{2}\right)\int_{-1}^1 d\xi\, e^{\frac{i}{2}\ell_{\mathrm{mirror}}(k_{j-1}-k_j)(\xi+1)}. \qquad \text{(C.17)}$$

According to Eq. (B.8), the SPA imposes the restriction $|k_1| = |k_2| = \dots = |k_p|$ on integrals contributing to the leading order; since in Eq. (C.16) the integration subdomain with $k_{F,R} < |k_j| < k_{F,L}$ is limited to $k_j > 0$, $\mathcal{M}_{\gamma_1\dots\gamma_p}$ is split into two independent contributions,

$$\mathcal{M}_{\gamma_1\dots\gamma_p} \sim \left(\frac{\ell_{\mathrm{mirror}}}{4\pi}\right)^p \int_{[-k_{F,R},k_{F,R}]^p} d^p k \sum_{\vec{a}\in\{0,1\}^{\otimes p}} \prod_{j=1}^p \left[\widetilde{\Xi}_{\gamma_j}^{a_{j-1}a_j}\left(k_{a_{j-1}},k_{a_j}\right)\Theta\left(k_{a_j}\right)\right]$$
$$+\left(\frac{\ell_{\mathrm{mirror}}}{4\pi}\right)^p \int_{[k_{F,R},k_{F,L}]^p} d^p k \prod_{j=1}^p \widetilde{\Xi}_{\gamma_j}^{00}(k_{j-1},k_j). \qquad \text{(C.18)}$$

The asymptotics of both integrals can be estimated using the SPA procedure explained before. When applied to the first integral (which corresponds to an equilibrium scenario), this procedure yields

$$\left(\frac{\ell_{\mathrm{mirror}}}{4\pi}\right)^p \int_{[-k_{F,R},k_{F,R}]^p} d^p k \sum_{\vec{a}\in\{0,1\}^{\otimes p}} \prod_{j=1}^p \left[\widetilde{\Xi}_{\gamma_j}^{a_{j-1}a_j}\left(k_{a_{j-1}},k_{a_j}\right)\Theta\left(k_{a_j}\right)\right]$$
$$\sim \ell_{\mathrm{mirror}} \frac{k_{F,R}}{\pi}\left\{\prod_{j=1}^p\left(1-e^{\frac{2\pi i\gamma_j}{n}}\right)+\prod_{j=1}^p\left(1+e^{\frac{-2\pi i\gamma_j}{n}}\right)\right\}, \qquad \text{(C.19)}$$

while for the second integral we find

$$\left(\frac{\ell_{\mathrm{mirror}}}{4\pi}\right)^p \int_{\left[k_{F,R},k_{F,L}\right]^p} d^p k \prod_{j=1}^{p} \widetilde{\Xi}^{00}_{\gamma_j}(k_{j-1},k_j) \sim \ell_{\mathrm{mirror}} \int_{k_{F,R}}^{k_{F,L}} \frac{dk}{2\pi} \left\{ \prod_{j=1}^{p} \left(1 - e^{\frac{2\pi i \gamma_j}{n}}\right) \right.$$
$$\left. + \prod_{j=1}^{p} \left[1 - e^{\frac{2\pi i \gamma_j}{n}} \mathcal{R}(k) + e^{\frac{-2\pi i \gamma_j}{n}} \mathcal{T}(k)\right] \right\}. \quad \text{(C.20)}$$

Together with Eq. (C.15), we then have

$$\mathrm{Tr}\left[\left\{\prod_{\gamma=-\frac{n-1}{2}}^{\frac{n-1}{2}} \left(\mathbb{I} - C_\gamma\right) - \mathbb{I}\right\}^s\right] \sim \ell_{\mathrm{mirror}} \int_{k_{F,R}}^{k_{F,L}} \frac{dk}{2\pi} \left\{\prod_{\gamma=-\frac{n-1}{2}}^{\frac{n-1}{2}} \left(e^{\frac{2\pi i\gamma}{n}}\mathcal{R}(k) - e^{\frac{-2\pi i\gamma}{n}}\mathcal{T}(k)\right) - 1\right\}^s$$
$$+ \Delta\ell_L \int_{k_{F,R}}^{k_{F,L}} \frac{dk}{2\pi} \left\{\prod_{\gamma=-\frac{n-1}{2}}^{\frac{n-1}{2}} \left(1 - \left(1 - e^{\frac{2\pi i\gamma}{n}}\right)\mathcal{R}(k)\right) - 1\right\}^s$$
$$+ \Delta\ell_R \int_{k_{F,R}}^{k_{F,L}} \frac{dk}{2\pi} \left\{\prod_{\gamma=-\frac{n-1}{2}}^{\frac{n-1}{2}} \left(1 - \left(1 + e^{\frac{-2\pi i\gamma}{n}}\right)\mathcal{T}(k)\right) - 1\right\}^s. \quad \text{(C.21)}$$

Using the decomposition of the polynomials $p_n$ and $\tilde{p}_n$ in Eqs. (C.6) and (C.7), we arrive at Eq. (23), a result that also captures the case $k_{F,L} < k_{F,R}$ (for which an equivalent derivation applies). By summing the series in Eq. (20) and taking the limit $n \to 1$, we then obtain the final result for the leading-order asymptotics of the fermionic negativity, given by Eq. (9).

# D  Additional numerical tests

Fig. 2 shows a comparison between our analytical results and numerical calculations of the different correlation measures – MI, CI and fermionic negativity – assuming a symmetric configuration of the subsystems ($\ell_L = \ell_R$ and $d_L = d_R$); Fig. 3 compares analytical and numerical results for the dependence of the MI and negativity on the positions relative to the scatterer of subsystems with fixed lengths. All of these results were computed for the single impurity model described in Sec. 3, for fixed values of the Fermi momenta and various values of the impurity energy $\epsilon_0$. In Fig. 4 we show similar comparisons, where now the impurity energy is fixed and results are plotted for various values of the bias, which is another parameter that influences the asymptotic scaling coefficients (for a bias that is small enough such that the scattering probabilities vary negligibly in $[k_-, k_+]$, the leading volume-law terms of the correlations measures are linear in the bias). Again, the good agreement of the analytical calculation with numerics is clearly evident.

The numerical calculations presented in Figs. 2–4 all rely on the direct diagonalization of two-point correlation matrices, through Eq. (14) (for the MI and CI) and Eq. (C.5) (for the negativity). The entries of these correlation matrices were computed in the limit $d_i/\ell_i \to \infty$ (with $d_L - d_R$ kept fixed), by discarding terms that vanish in this limit according to the Riemann-Lebesgue lemma, as explained in Appendix A.

In Fig. 5 we demonstrate that the omission of these terms from the correlation matrices indeed captures the $d_i/\ell_i \to \infty$ limit of the correlation measures themselves. For the symmetric case $\ell_L = \ell_R = \ell$ and $d_L = d_R = d$, we let $\mathcal{I}^{(d)}_{A_L:A_R}$ and $\mathcal{E}^{(d)}$ denote the MI and negativity, respectively, that were numerically calculated using correlation matrices with entries given

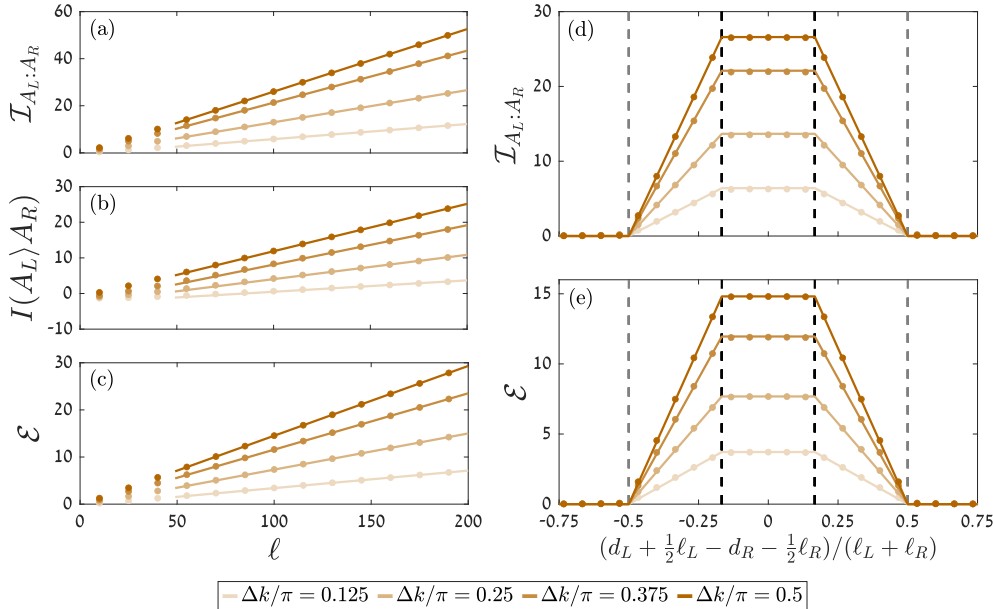

Figure 4: The single impurity model: Scaling of (a) the mutual information, (b) the coherent information, and (c) the fermionic negativity between subsystems $A_L$ and $A_R$ for the symmetric case $\ell_L = \ell_R = \ell$ and $d_L = d_R$. (d) The mutual information and (e) the fermionic negativity as a function of $d_L - d_R$, when fixing $\ell_L = 100$ and $\ell_R = 200$; letting $\bar{A}_L = \{m \,|\, -m \in A_L\}$ denote the mirror image of $A_L$, black dashed vertical lines mark the boundaries of the domain where $\bar{A}_L \subset A_R$, while gray dashed vertical lines mark the boundaries of the domain where $\bar{A}_L \cap A_R \neq \phi$. In all the panels, results are computed in the limit $d_i \gg \ell_i$; analytical results (lines) are compared to numerical results (dots) for different values of the bias $\Delta k = k_{F,L} - k_{F,R}$, with the lower Fermi momentum fixed at $k_{F,R} = \pi/2$, and the impurity energy fixed at $\epsilon_0 = \eta$.

by Eq. (12), while $\mathcal{I}_{A_L:A_R}^{(\infty)}$ and $\mathcal{E}^{(\infty)}$ stand, respectively, for the MI and negativity that were numerically calculated using correlation matrices with entries given by Eq. (A.3).

Indeed, the results in Fig. 5, numerically computed for the single impurity model, indicate that $\mathcal{I}_{A_L:A_R}^{(d)} \to \mathcal{I}_{A_L:A_R}^{(\infty)}$ and $\mathcal{E}^{(d)} \to \mathcal{E}^{(\infty)}$ as $d/\ell \to \infty$. As they converge toward this limit, the correlation measures exhibit Friedel oscillations, a behavior that was previously observed for the entanglement entropy of a single subsystem of contiguous sites [28]. As in that case, the difference between the average over these oscillations and the $d \to \infty$ limit decays according to a $\propto 1/d^2$ power law, while the amplitude of the oscillations decays according to a $\propto 1/d$ power law.

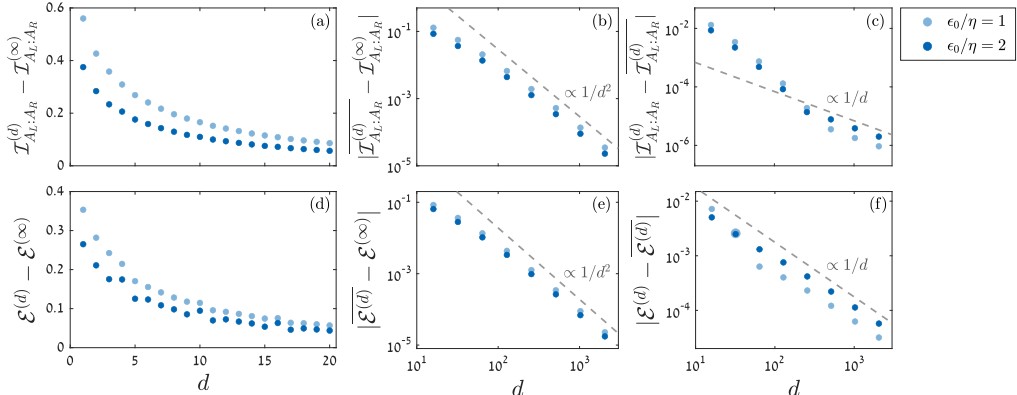

Figure 5: The single impurity model: Comparison between numerical calculations of correlation measures for the symmetric case with $\ell_L = \ell_R = \ell = 50$ and $d_L = d_R = d$, for two values of the impurity energy $\epsilon_0$ and with the Fermi momenta fixed at $k_{F,R} = \pi/2$ and $k_{F,L} = 2\pi/3$. (a) The difference between $\mathcal{I}^{(d)}_{A_L:A_R}$, the MI computed using the full expressions for the correlation matrices (Eq. (12)), and $\mathcal{I}^{(\infty)}_{A_L:A_R}$, the MI computed from correlation matrices where entries were taken to the limit $d \to \infty$ (Eq. (A.3)), as a function of $d$. (b) The deviation of $\overline{\mathcal{I}^{(d)}_{A_L:A_R}}$, the average of $\mathcal{I}^{(d)}_{A_L:A_R}$ over Friedel oscillations, from $\mathcal{I}^{(\infty)}_{A_L:A_R}$; the dashed gray line emphasizes that, for $d \gg \ell$, the deviation approaches a $\propto 1/d^2$ power-law behavior. (c) The amplitude $\left| \mathcal{I}^{(d)}_{A_L:A_R} - \overline{\mathcal{I}^{(d)}_{A_L:A_R}} \right|$ of the oscillations in the MI; the dashed gray line emphasizes that, for $d \gg \ell$, the amplitude approaches a $\propto 1/d$ power-law behavior. The bottom panels (d)–(f) present a similar analysis for the fermionic negativity $\mathcal{E}$.

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
