# Peer review of "Extensive long-range entanglement in a nonequilibrium steady state"

_SciPost Physics, doi:SciPost Phys. 15, 134 (2023)_

## Round 1 · Referee Report · Anonymous · 2023-6-18

Strengths

Well-written paper supported by several technical details about the derivation of the main results.

Report

In this paper, the authors study the effect of a localised impurity in a tight-binding chain. Starting from the expression for the correlation matrix of this model, they find its stationary value exploiting the stationary phase approximation. This allows them to compute three entanglement measures, which are the (Renyi) mutual information, the (Renyi) negativity and the coherent information.
I find the paper is very well written. Fig. 1 makes clear the setup the authors consider, and the presence of very detailed appendices have two advantages: they do not interrupt the flow in the main text, such that the message is not hidden behind the technical computations, but at the same time they provide all the details to reproduce the results. For this reason, I recommend the publication of the manuscript. Nevertheless, a short list of more specific comments/questions follows.

Requested changes

1) The analysis of this manuscript can be also done for a generic fermionic Gaussian state (i.e. where also pairing terms like $c^{\dagger}_{i}c^{\dagger}_{i+1}$ are in the Hamiltonian (4))? Do you think that, even starting from a short-range entangled state, the quantum correlations can be enhanced by the presence of the scattering region?

2) When the authors state that this model describes a class of nonequilibrium steady states, do open systems with gain/loss terms enter into this class?

3) In Eq. B7 there should be $F(\vec{\sigma},\vec{\sigma})$?

4) In the bottom right panel of Fig. 5 there is one (light blue) point missing. Why in the amplitude of the oscillations appears a crossing between the light and dark blue points, while in the difference between the average over the oscillations it does not?

5) Can the authors write explicitly the relation between the Fermi momenta $k_{F,L/R}$ and the chemical potential $\mu_{L,R}$?

  • validity: high
  • significance: good
  • originality: good
  • clarity: high
  • formatting: good
  • grammar: excellent

Author:  Shachar Fraenkel  on 2023-08-10  [id 3893]

(in reply to Report 1 on 2023-06-18)

We thank the Referee for their diligent reading of our manuscript, and for recommending its publication in SciPost Physics. Below we list our responses to the specific points raised by the Referee.

1) The Referee asks: "The analysis of this manuscript can be also done for a generic fermionic Gaussian state (i.e. where also pairing terms like $c_i^{\dagger}c_{i+1}^{\dagger}$ are in the Hamiltonian (4))? Do you think that, even starting from a short-range entangled state, the quantum correlations can be enhanced by the presence of the scattering region?" It is not completely clear to us to what extent the general entanglement-generating mechanism that we describe survives in the presence of pairing terms in the bulk Hamiltonian, given that these terms break charge conservation. This problem might require a modified definition of the chemical potential, and in any case seems to merit a separate and careful consideration. We now mention this as an additional future direction for research in the Conclusions section.

2) The Referee asks: "When the authors state that this model describes a class of nonequilibrium steady states, do open systems with gain/loss terms enter into this class?" Indeed, we expect that our qualitative observations will apply also to the case where the impurity induces particle gain and loss. This will simply amount to a modification of the associated scattering matrix such that the sum of reflection and transmission probabilities will be smaller than 1, while the essential coherence between the reflected and transmitted parts will still be present (see Ref. [29] which discusses a specific instance of such a scenario). We have added a mention of this scenario to the Conclusions section.

3) The Referee asks: "In Eq. B7 there should be $F(\vec{\sigma},\vec{\sigma})$?" We thank the Referee for pointing out the typo, which has been corrected.

4) The Referee asks: "In the bottom right panel of Fig. 5 there is one (light blue) point missing. Why in the amplitude of the oscillations appears a crossing between the light and dark blue points, while in the difference between the average over the oscillations it does not?" The light blue point to which the Referee refers was hiding behind the dark blue point. We enlarged it to make it more visible. As for the Referee's question regarding the trends seen in Fig. 5, we note that their detailed analysis is beyond the scope of our manuscript. Our intention was to very generally understand the size of the corrections to our asymptotic result and to see when they can be neglected, rather than to provide a thorough description of their behavior.

5) The Referee asks: "Can the authors write explicitly the relation between the Fermi momenta $k_{F,L/R}$ and the chemical potential $\mu_{L,R}$?" We added a sentence to that effect near the end of Section 2.

---

## Round 1 · Referee Report · Anonymous · 2023-6-23

Report

The authors study entanglement properties in a chain of free fermions, in the presence of a scattering region located in the middle of the chain. In particular, they focus on nonequilibrium steady states that emerge due to a density bias on the two sides. They find that, for two disjoint intervals located on opposite sides of the scatterer, both the mutual information (which is a measure of the total correlations) as well as the entanglement negativity (which is a genuine entanglement measure) scale linearly with the number of sites shared between one interval and the mirror image of the other. This holds true independently of the distance from the scattering region, i.e. one has a special kind of long-range entanglement in the NESS with a volume-law scaling. This behaviour can be interpreted in terms of entangled pairs of partially transmitted and reflected waves, that carry the entanglement over arbitrary long distances. Consequently, the MI and negativity are functions of the transmission coefficient only. The mechanism at hand was identified before in [21] and studied for adjacent intervals, which is now generalized to subsystems in arbitrary but similar distances from the center. The proof uses saddle-point techniques that were first introduced for the study of entropy evolution in global quench protocols.

I believe that this is an interesting work with sound results, which deserves publication. The paper is well written, with the main text containing only the main steps of the derivation, whereas all the technicalities are relegated to several appendices. This structure makes the manuscript very readable.

Requested changes

- I would have been interested in the results for the subleading logarithmic terms, but apparently this will be presented in a separate publication.

- Minor issue: in Appendix A (bottom of page 10) the quantity $\alpha_{j,m}$ is not defined. Does it simply denote the factor $j-m$ or $j+m$ ?

  • validity: -
  • significance: -
  • originality: -
  • clarity: -
  • formatting: -
  • grammar: -

Author:  Shachar Fraenkel  on 2023-08-10  [id 3894]

(in reply to Report 2 on 2023-06-23)

We greatly thank the Referee for their thorough reading of our manuscript, and for recommending it for publication. Below we list our responses to the specific issues raised by the Referee.

1) The Referee writes: "I would have been interested in the results for the subleading logarithmic terms, but apparently this will be presented in a separate publication." We may point out that the explicit expressions for the subleading logarithmic terms, along with their derivation, already appeared in the original arXiv posting of our results (arXiv:2205.12991), and can be seen there. After posting that original version, we decided to split it into two manuscripts, in order to improve the clarity and readability of the work, and given that the two calculations require quite different methodologies. The separate manuscript that discusses the logarithmic terms is currently in preparation and will be uploaded to arXiv in the near future. We of course thank the Referee for their interest in this aspect of our work.

2) The Referee writes: "In Appendix A (bottom of page 10) the quantity $\alpha_{j,m}$ is not defined. Does it simply denote the factor $j-m$ or $j+m$?" Indeed this was our intention, and in the revised version we now state it explicitly.

---

## Round 2 · Author Response

Dear Editor,

We hereby submit the requested revised version of our manuscript. We would like to thank the Referees for their valuable comments on our manuscript, and for recommending it for publication in SciPost Physics. In each of our replies to the Referees we detail the specific changes that we have made following their comments. These modifications are also marked in red in the revised version. With this, we now believe that our manuscript is ready for publication.

Sincerely,
Shachar Fraenkel and Moshe Goldstein

---

## Round 2 · List of Changes

- Short remarks added to Section 2, Section 5 and Appendix A (all marked in red).
- Typo fixed in Eq. (B7).
- A data point in panel (f) of Fig. 5 that was originally hidden has been enlarged to improve its visibility.

---

## Editorial Decision

published